# Amplifying metabolic profiling of extracellular vesicle dynamics with ACTIVITY

Ru-Jia Yu[1], Wei-Yi Ma[1], Han-Yang Xiao[1], Ya-Wei Zhang[1], Wen-Bin Gong [2], Zhen-Fei Yu[1], Kai-Liang Wei[1], Kuo-Ran Xing [3], Xu Wang[4], Hou-Juan Zhu [5], Lian-Hui Wang [1] & Xian-Guang Ding [1] ✉

Extracellular vesicles (EVs) are emerging as promising circulating biomarkers for liquid biopsy due to their abundant molecular information, such as proteins, nucleic acids, and metabolites. However, the metabolic profiling of EVs remains largely unexplored, much less exploiting their intrinsic metabolic features for disease diagnosis. In this study, we demonstrate that the metabolic-related inducible nitric oxide synthase (iNOS) activity of macrophage-derived EVs serves as an effective biomarker for phenotypic profiling and further evaluating lung inflammation. By integrating a cascade amplification strategy that combines the biocatalysis of EV iNOS activity with the electrocatalysis of defective tungsten disulfide quantum dots ($WS_2$ QDs), we develop an ACTIVITY (Amplified Cascade-catalysis TestIng for VesIcular meTabolic activitY) method for rapid assaying of metabolically active EVs. When applied to bronchoalveolar lavage fluid samples, this activity-based EV profiling differentiates pneumonia patients from healthy controls and further facilitates the monitoring of disease treatment, suggesting the potential of EV metabolic activity for diagnostics.

Extracellular vesicles (EVs) are membranous nano-entities secreted by nearly all cell types and found in various body fluids (e.g., blood, saliva, urine, tears, cerebrospinal fluid, lavage fluid, etc.)[1]. Their secretion is a pervasive and persistent process observed across most organs and tissues, occurring under both normal and pathological conditions. These EVs contain abundant molecular information, such as proteins, nucleic acids, and metabolites that indicate physiological and pathological states of the body. They have thus been recognized as a promising circulating biomarker for liquid biopsy[2]. For example, target protein profiling of blood EVs has been employed for the clinical diagnosis of cancer[3–5], tuberculosis[6], neurodegenerative disease[7,8]. Identification of target nucleic acids[9,10] and even the intricate glycans[11]

in tear or blood EVs has been utilized for disease diagnosis and monitoring. In fact, metabolites participate as the final downstream products of biological activities, serving as the fundamental immediate indicator for abnormality in the microenvironment. In recent years, it has been reported that EVs sometimes exhibit enzymatic activities and behave as autonomous metabolic units, which holds promise to affect the physiology of their microenvironment[12]. However, whether these metabolically active EVs can serve as an independent biomarker, capable of disease interpretation, is still unknown.

Macrophages, as heterogeneous innate immune cells, display distinct phenotypes in response to different microenvironmental stimuli, and these differences are reflected in their intrinsic metabolic

[1]State Key Laboratory of Flexible Electronics (LoFE) and Jiangsu Key Laboratory of Smart Biomaterials and Theranostic Technology, Institute of Advanced Materials (IAM), Nanjing University of Posts and Telecommunications, Nanjing, China. [2]School of Physics and Energy, Xuzhou University of Technology, Xuzhou, China. [3]Department of Chemical and Biomolecular Engineering, National University of Singapore, Singapore, Singapore. [4]The First Affiliated Hospital of Soochow University, Department of Central Intensive Care Unit of Anesthesiology, Suzhou, China. [5]A*STAR (Agency for Science, Technology and Research), Singapore, Singapore. ✉e-mail: iamxgding@njupt.edu.cn

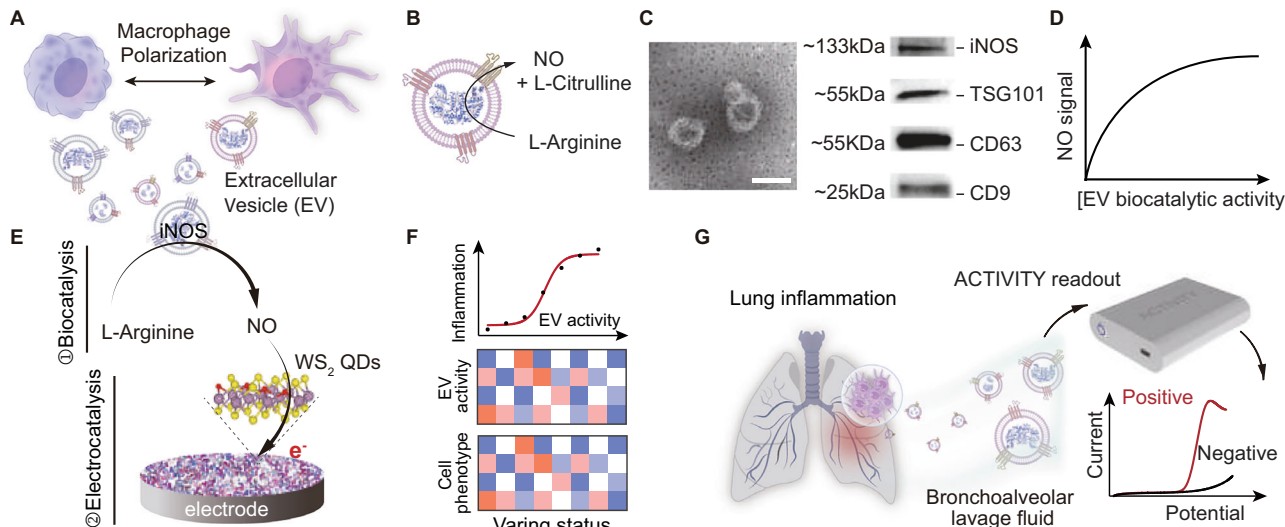

**Fig. 1 | ACTIVITY-based analysis of EV metabolic dynamics. A** Macrophages of different phenotypes secrete extracellular vesicles (EVs) with varying metabolic activity. Photo credit: Han-Yang Xiao. **B** The iNOS (inducible nitric oxide synthase) activity of EVs catalyzes the conversion of L-arginine into nitric oxide (NO). Photo credit: Han-Yang Xiao. **C** TEM image and Western blot analysis of macrophage-derived EVs. scale bar, 100 nm. The experiment was independently repeated three times with similar results. **D** The generation of NO signals serves as an indicator of the intrinsic catalytic activity of EVs. **E** A cascade-catalytic amplification for the

metabolic activity profiling of macrophage-derived EVs with varying metabolic activity. Photo credit: Ru-Jia Yu. **F** Metabolic profiling of the macrophage-derived EVs facilitates the identification of macrophage phenotype, indicating the inflammation in the relevant organ. **G** The ACTIVITY method was further used to detect lung macrophage-derived EVs in bronchoalveolar lavage fluid samples, allowing the discrimination of lung inflammation. Photo credit: Ya-Wei Zhang. Source data are provided as a Source Data file.

profiles[13]. They are generally categorized into the pro-inflammatory phenotype and the anti-inflammatory phenotype, with the polarization between these two phenotypes being strongly associated with the occurrence and progression of inflammation[14,15]. Specifically, M1 macrophages are characterized by elevated levels of inducible nitric oxide synthase (iNOS), a metabolic enzyme synthesizing nitric oxide (NO) from L-arginine[16]. In the inflammatory conditions affecting internal organs such as the lung, direct isolation of the macrophages for analysis is challenging, and conventional diagnostic methods generally involve imaging techniques such as magnetic resonance imaging and chest computed tomography[17]. Recent studies have highlighted the role of lung macrophage-derived EVs in intercellular crosstalk during pneumonia, suggesting their potential as disease biomarkers[18,19]. While existing studies have mainly focused on genomic and proteomic analysis, investigations into the metabolic enzyme iNOS have largely focused on its vesicular protein levels rather than its intrinsic biocatalytic activity. Therefore, directly assessing the iNOS activity of macrophage-derived EVs, which reflects the dynamic metabolic state of the parental macrophages, represents a promising diagnostic approach for evaluating inflammation in internal organs.

The development of biological assays is highly dependent on signal amplification strategies, which dominate the sensitivity of analytical methods[20]. Amongst these, enzyme-based biocatalysis, as well as nanoparticle-based chemical catalysis, have attracted increasing attention. Recent advancements in atomic defect engineering of nanoparticles have further enhanced their catalytic performance when used as nanocatalysts[21,22]. In typical electrochemical biosensors, incorporating bioenzymes or nanoparticles onto the electrode surface serves as an effective signal amplification, allows for highly sensitive and selective detection[23]. Herein, we transform the iNOS activity of EVs into a biocatalytic amplification, using the enzymatic product NO as an electrochemical redox probe for metabolically active EV detection. Meanwhile, a synergistic electrocatalysis-based amplification is further integrated by modifying the electrode with highly defective $WS_2$ QDs for electrocatalytic oxidation of NO. By integrating this cascade signal amplification strategy, we develop an ACTIVITY (Amplified Cascade-

catalysis TestIng for VesIcular meTabolic activitY) method for highly sensitive detection of iNOS biocatalytic activity of EV (EV-iNOS) (Fig. 1). It enables rapid profiling of the metabolically active EVs from bronchoalveolar lavage fluid (BALF), which reflect macrophage phenotypes and are capable of indicating inflammation states of the lung. As a result, this metabolic activity of EVs is proposed as an independent metabolic unit as an effective liquid biopsy marker for pneumonia diagnosis. Compared to the conventional ELISA method for vesicular protein detection, this functional measurement of EV activity improves the differentiation of pneumonia patients and enables the evaluation of clinical disease treatment. This strategy proposed here allows for sensitive and rapid assaying of metabolically active EVs, providing diverse applications in inflammation diagnosis and treatment monitoring.

## Results and discussion
### Design and evaluation of ACTIVITY probes
The $WS_2$ QDs were synthesized using a bottom-up method through the one-pot reaction between $WO_3$ and $NaS_2$ in an aqueous solution, with glutathione acting as the surfactant (Fig. 2A). Hydrogen peroxide was further employed to introduce S defects in these QDs through mild chemical etching[24]. By adjusting the etching process, $WS_2$ QDs with different degrees of defects were obtained, showing as $WS_2$-$D_L$ (low defects), $WS_2$ $D_M$ (medium defects), and $WS_2$-$D_H$ (high defects). These colloidal QDs showed good dispersibility and stability in aqueous solutions, with the sample color changing from pale green to cyan and blue as defect density increased (Supplementary Fig. 1A). Corresponding transmission electron microscopy (TEM) images revealed that all three defective QDs showed homogeneous size distributions, with a similar particle size of 4 nm (Fig. 2B–D). The UV-Vis absorption spectra of these defective $WS_2$ QDs exhibited strong absorption at 400–500 nm without multiple excitonic peaks (Supplementary Fig. 1B), indicating a strong quantum size effect for QDs. X-ray diffraction (XRD) patterns of these $WS_2$ QDs showed broad diffraction peaks, supporting the nanoscale dimension of these nanoparticles (Supplementary Fig. 2). Major diffraction peaks indexed to (100) and

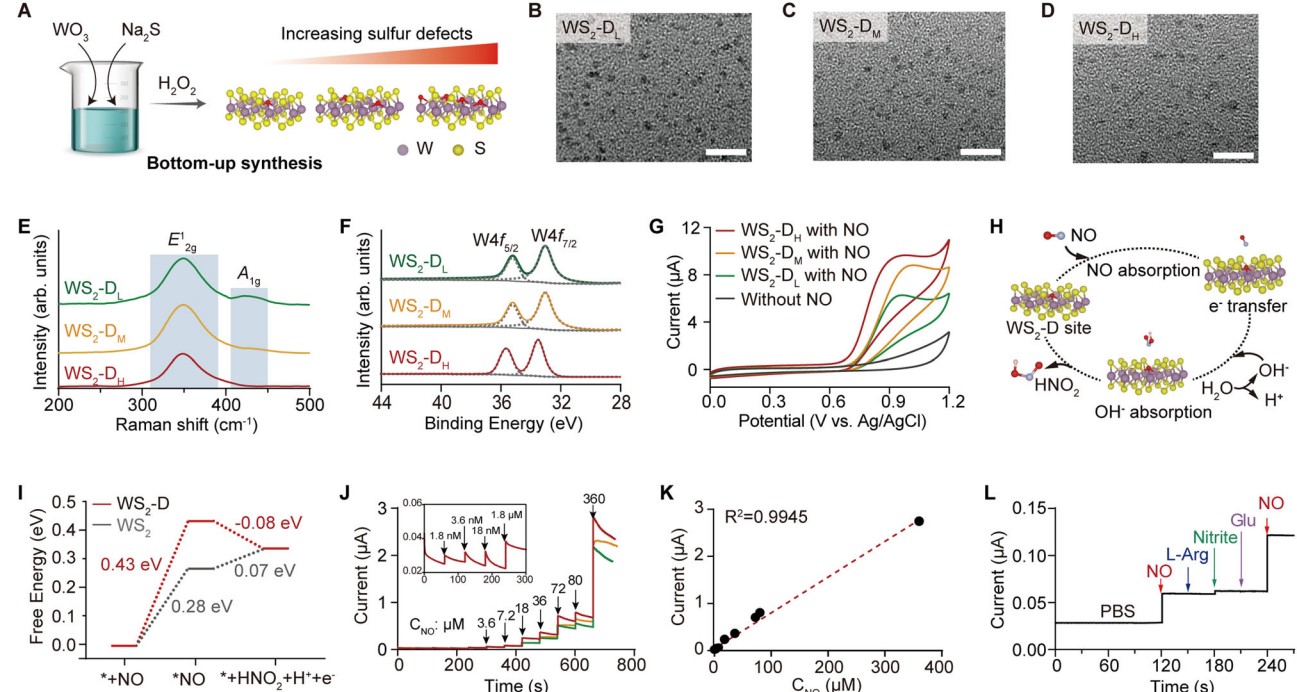

**Fig. 2 | Design and evaluation of ACTIVITY probes. A** The synthesis of defective WS₂ QDs via a one-pot reaction. Photo credit: Ya-Wei Zhang. TEM images of WS₂-D$_L$ (**B**), WS₂-D$_M$ (**C**), and WS₂-D$_H$ QDs (**D**). Scale bar, 50 nm. The experiment was independently repeated three times with similar results. **E** Raman and **F** XPS spectra of these three kinds of defective WS₂ QDs. **G** Cyclic voltammograms of different defective WS₂ QDs modified electrodes in PBS electrolyte with 0.18 mM NO. **H** Structure illustration of the electrocatalytic mechanism of defective WS₂ QDs for NO oxidation. **I** Gibbs free energy diagram for the pathway of NO oxidation catalyzed by WS₂ QDs. The denotation of * represents the active site in WS₂ QDs. **J** Amperometric responses of WS₂-D$_H$ (red curve), WS₂-D$_M$ (orange curve), and WS₂-D$_L$ (green curve) QDs modified electrodes upon successive addition of NO at 0.8 V. The inset shows the enlarged amperometric response of WS₂-D$_H$ QDs modified electrode from 0 to 300 s. **K** Calibration curve of the current response of the WS₂-D$_H$ QDs modified electrode to NO concentration. **L** Amperometric response of the WS₂-D$_H$ QDs modified electrode toward interfering chemicals, including L-arginine, nitrite, glucose, and NO. Source data are provided as a Source Data file.

(110) shifted toward higher angles were observed with elevated defects on the WS₂ QDs surface. This shift was mainly attributed to the gradual enlargement of the lattice constants, which occurred as the larger sulfur atom was replaced by the smaller oxygen atom. Figure 2E demonstrated the typical Raman spectra of these WS₂ QDs with excitation wavelength at 532 nm, showing two characteristic peaks at ~350 cm⁻¹ and ~420 cm⁻¹ for the $E^1_{2g}$ and $A_{1g}$ modes, respectively[25]. The former corresponded to the in-plane motion of W and S atoms, whereas the latter revealed the vibration of S atoms in the out-of-plane orientation. The observed Raman peaks demonstrated width broadening and intensity quenching with increased defect densities, which agrees well with previous literature[26]. High-resolution X-ray photoelectron spectroscopy (XPS) was further performed to evaluate the constituent elements. Characteristic doublet peaks for W $4f_{5/2}$ and W $4f_{7/2}$ were observed at binding energy of ~35 eV, indicating the W (IV) oxidation state in these QDs (Fig. 2F). The increasing S defects could also be confirmed by the shifted peak towards higher binding energy. In short, we hereby present a simple and controllable method for defect engineering at the WS₂ QDs surface.

The defect-modulated electrocatalytic performance of WS₂ QDs toward NO oxidation was first evaluated by cyclic voltammetry (CV). These three types of QDs, including WS₂-D$_L$, WS₂ D$_M$, and WS₂-D$_H$, were used to modify glassy carbon electrodes, serving as the working electrodes. Nafion-assisted WS₂ QDs modification method was utilized to provide an electrode surface with high stability and selectivity. The amount of three samples of WS₂ QDs for electrode coating was controlled to be consistent. Within a standard three-electrode system, all three defective WS₂ QDs showed clear catalytic activities toward the electrooxidation of NO, with the NO oxidation peak observed at ~0.8 V (Fig. 2G). Clearly, the oxidative current was greatly enhanced

compared with that of the bare electrode (Supplementary Fig. 3). Density functional theory (DFT) calculations were performed to explore the defect-modulated electrocatalytic mechanism (see Supplementary Fig. 4 for the calculated density of states of these QDs). It is generally acknowledged that NO oxidation mainly involves two steps: the adsorption of NO on the catalyst, and its subsequent oxidation with H₂O, forming HNO₂ at the catalyst surface (Fig. 2H)[27]. The Gibbs free energy profile along the electrooxidation pathway showed that the Gibbs free energy change for NO adsorption was 0.43 eV for the defective WS₂ QDs, slightly larger than that of non-defective WS₂ QDs (0.28 eV), indicating a thermodynamically unfavorable process (Fig. 2I). However, considering the change in Gibbs free energy for the total reaction remains fixed, the electron transfer step is favorable for defective WS₂ QDs. The theoretical results demonstrated that the defects in WS₂ QDs enhance the electrocatalytic performance towards NO oxidation, supporting this method for highly sensitive NO detection.

The time-dependent current responses with successive NO additions were obtained by amperometric measurement, with stirring performed after each addition to ensure solution homogeneity. As demonstrated in Fig. 2J, these three defective WS₂ QDs modified electrodes showed rapid responses to NO addition, with the WS₂-DH QDs exhibiting the highest responsive current signals at the nanomolar level. This could also be explained by the CV curves in Fig. 2G, where the NO oxidation potential at WS₂-D$_H$ was more negative, and its peak current was much higher than those of WS₂-D$_L$ and WS₂ D$_M$, indicating the highest catalytic activity toward NO oxidation. Also, these modified electrodes exhibited better electrocatalytic performance than the generally used carbon-based electrodes such as reduced graphene oxide-modified electrode (Supplementary Fig. 5). The stability of WS₂-

DH QDs modified electrode was evaluated by multiple CV measurements, which showed a stable NO oxidation peak with the relative standard deviation (RSD) about 3.3% over 12-h duration at 2-h intervals (Supplementary Fig. 6). The WS$_2$-D$_H$ QDs-based electrochemical sensor showed a linear relationship between the current response and NO concentrations, enabling detection at the nanomolar level (Fig. 2K). The selectivity of this method was further evaluated by introducing relevant interferences in the biological system, including L-arginine, nitrite, and glucose. As shown in the amperometric response, the addition of these interfering substances at concentrations of 0.1 mM resulted in negligible contributions to the oxidation current, with only the addition of NO (5 μM) showing a clear current response (Fig. 2L). The cation-selective Nafion film, as well as the high electrocatalytic WS$_2$-D$_H$ QDs on the electrode surface, contributed to the high selectivity of this method. These results demonstrate that this WS$_2$-DH QDs-based sensor exhibits excellent performance in NO quantification, showing both high sensitivity and selectivity. It could be further integrated to detect NO-generating bioentities such as iNOS and metabolically active EVs that exhibit iNOS activity.

## ACTIVITY for EV metabolic activity analysis

EVs derived from RAW 264.7 cells were isolated from the cell culture supernatant via the ultracentrifuge method. The isolated EVs showed an average diameter of ~120 nm (nanoparticle tracking analysis in Fig. 3A) with a typical discoid morphology (TEM image in Fig. 1C). Western blot analysis confirmed the presence of iNOS protein in these macrophage-derived EVs, together with the characteristic EV transmembrane markers including CD9, CD63 and TSG101 (Fig. 1C). To explore the exact location of the iNOS protein within EVs, we quantified both iNOS protein content and its biocatalytic activity in (i) lysed and (ii) intact EVs. As shown in Fig. 3B, ELISA measurements revealed higher levels of iNOS protein in lysed EVs compared with intact EVs, indicating limited accessibility of iNOS without membrane disruption. Meanwhile, iNOS biocatalytic activity was assessed by measuring NO production using a NO fluorescent probe, diaminofluorescein-FM in the presence of enzyme substrates. Lysed EVs exhibited a clear, time-dependent increase in NO production, whereas intact EVs produced very little NO (Fig. 3C). These results demonstrate that the disruption of EVs structure is essential for iNOS sensing, strongly suggesting that iNOS is localized within the lumen of EVs rather than on the surface. For this reason, iNOS exhibited considerable stability during EV storage, with its biocatalytic activity remained nearly unchanged over 42 days at −80 °C (Fig. 3D). This enhanced stability is probably attributed to the inside localization of iNOS, where lipid encapsulation provides effective protection during the long-term storage.

We further employed this defect-enhanced method to assess the iNOS activity of macrophage-derived EVs (referred to as metabolic EVs-iNOS) and test whether this metabolic signature could be used to characterize macrophage polarization (Fig. 3E). During the amperometric recording, pure EV lysate showed negligible interference without adding enzyme substrates (Supplementary Fig. 7). To further validate the iNOS activity of EVs, enzyme substrates were introduced to enable NO generation for electrochemical readout. As shown in Supplementary Fig. 8, the presence of iNOS containing EVs, but not the heat-inactivated ones, produced a clear current response comparable to that observed for NO. It suggested that the oxidative current was largely due to the intrinsic iNOS activity of the EVs, rather than the original NO leakage. The analytical capacities of this method for EVs-iNOS were further compared to a conventional ELISA method by using anti-iNOS antibodies for protein quantification. Independent detections with varying EV concentrations revealed that the ACTIVITY and ELISA methods achieved a limit of detection of ~1000 and ~10$^6$ EVs, respectively (Fig. 3F). Further EV-iNOS measurement was conducted using the NO fluorescent probe, diaminofluorescein-FM diacetate, for EV dilution analysis, yielding a limit of detection of ~10$^5$ EVs (Fig. 3G). It

demonstrated that the proposed ACTIVITY method exhibits much higher sensitivity compared to the conventional methods. This improvement is attributed to the high electrocatalytic performance of defective WS$_2$ QDs in NO detection and the effective signal amplification through the integrated enzymatic process of the EVs. Meanwhile, this ACTIVITY method is unlikely to be affected by the antibody capture efficiency, thereby contributing to the improved sensitivity. The stability test of the ACTIVITY method showed less than a 15% current drop over 14 days at an EV concentration of 5×10$^4$, indicating that the WS$_2$-D$_H$ QDs-coated electrode remained stable electrocatalytic activity throughout the storage period (Fig. 3H). These results suggest that this ACTIVITY method offers a superior option for detecting metabolically activity of EV, showing both high sensitivity and stability.

Next, we investigated the capability of this ACTIVITY approach for quantitatively characterizing macrophage polarization. This method was employed to analyze EVs sourced from anti-inflammatory and pro-inflammatory macrophages separately, as well as the lysis of parental cells. Supplementary Fig. 9 demonstrated that both EVs and cell lysates show distinct iNOS metabolic differentiation between anti-inflammatory and pro-inflammatory phenotypes, with notably higher levels of metabolic EVs-iNOS in the pro-inflammatory polarization macrophages. To further evaluate the capability of this ACTIVITY method for quantifying macrophage polarization, we used inflammatory macrophage models stimulated with lipopolysaccharide (LPS) to induce pro-inflammatory macrophages[28]. RAW 264.7 macrophages pretreated with IL-4 to establish an anti-inflammatory starting state were then incubated with LPS for varying durations. Western blot analysis verified the iNOS overexpression in the pro-inflammatory macrophages (Supplementary Fig. 10). Given that CD86 and CD206 are characteristic markers of pro-inflammatory and anti-inflammatory macrophages, respectively, the intracellular CD86/CD206 ratio obtained from flow cytometry was used as a referential indicator for macrophage polarization (Supplementary Fig. 11). As shown in Fig. 3I, the CD86/CD206 ratio increased with LPS incubation time, indicating that the macrophages polarized towards the pro-inflammatory phenotype. The corresponding metabolic EVs-iNOS levels from LPS-treated macrophages were measured by isolating EVs from the culture supernatant and lysed for ACTIVITY assay. Figure 3J showed that the relative current responses, representing metabolic EVs-iNOS levels, increased with longer LPS incubation. Similarly, BLZ945, which is known to suppress anti-inflammatory polarization, was used to regulate macrophage polarization (Fig. 3K). The metabolic EVs-iNOS levels in Fig. 3L showed a consistent increased tendency over more extended BLZ945 incubation periods. Briefly, both models demonstrated that the metabolic activity of EVs was proportionally correlated with pro-inflammatory polarization of macrophages. Figure 3M shows a high correlation (r = 0.9865) between this metabolic EVs-iNOS and the polarization level obtained from cellular CD86/CD206, indicating that metabolic EVs-iNOS could serve as an effective cell-free indicator for quantifying macrophage polarization. Additionally, intracellular iNOS levels in whole-cell lysates were measured, verifying increased protein expression associated with anti-inflammatory polarization (Supplementary Fig. 12, 13). To verify the essential role of metabolic EVs-iNOS in characterizing macrophage polarization, an iNOS gene knockout RAW 264.7 cell model was constructed using the CRISPR/Cas9 system (Supplementary Figs. 14, 15). As shown in Fig. 3N, EVs derived from iNOS knockout cells treated either with LPS or IL-4 exhibit consistently low metabolic EVs-iNOS levels, where the current was normalized to normal RAW264.7 derived-EVs. Primary macrophages collected from mouse peritoneal were employed for further validation, showing a clear differentiation in the metabolic EVs-iNOS levels between the different polarized cells (Fig. 3O). Notably, this ACTIVITY-based readout can be easily implemented using portable electrochemical devices. To facilitate its broad applicability, we integrated this technology into

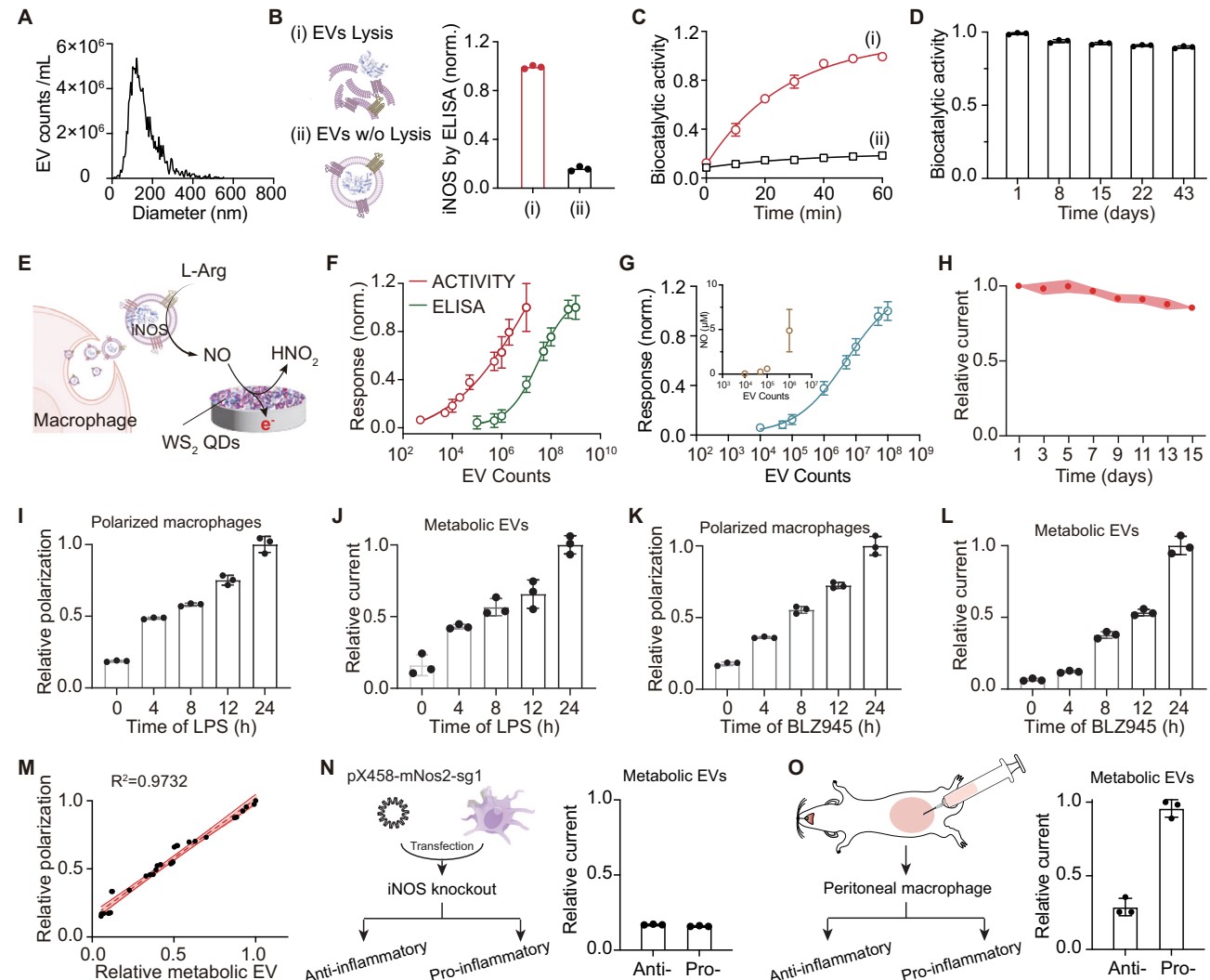

**Fig. 3 | ACTIVITY for EV metabolic activity analysis. A** Particle size distribution of macrophage-derived EVs. **B** Lysis-triggered protein detection. iNOS concentrations in (i) lysed and (ii) intact EVs samples determined by ELISA ($n = 3$ independent experiments). Photo credit: Ru-Jia Yu. **C** Normalized biocatalytic activity in (i) lysed and (ii) intact EVs ($n = 3$ independent experiments). **D** Biocatalytic stability of EVs during storage at $-80\,°C$ ($n = 3$ independent experiments). Relative activity on a given day was normalized to freshly acquired EVs. **E** Schematic diagram of the ACTIVITY method via a two-step mechanism. This includes EV-catalyzed L-Arg oxidation, yielding NO to be electrocatalytically oxidized by defective $WS_2$ QDs on the electrode. Photo credit: Ru-Jia Yu. **F** Comparison of detection sensitivity between the proposed ACTIVITY method and ELISA ($n = 3$ independent experiments). Data are presented as mean ± 95% CI. **G** Normalized fluorescent intensities of diaminofluorescein-FM diacetate in response to EV concentration ($n = 3$ independent experiments). The inset shows corresponding NO concentrations, obtained by correlating the fluorescent intensities to a standard curve. **H** Stability of the ACTIVITY method for the detection of metabolically active EVs ($n = 3$ independent experiments). Relative current on a given day was normalized to the original amperometric response immediately after electrode preparation. **I** LPS modulated polarization of macrophages obtained by flow cytometric analysis using anti-CD86 and anti-CD206 ($n = 3$ independent experiments). **J** ACTIVITY readout of metabolic EVs-iNOS levels in the supernatant by incubating macrophages with LPS over time ($n = 3$ independent experiments). **K** BLZ945 modulated polarization of macrophages obtained from flow cytometric analysis ($n = 3$ independent experiments). **L** ACTIVITY readout of metabolic EVs-iNOS levels by incubating macrophages with BLZ945 over time ($n = 3$ independent experiments). All metabolic EVs-iNOS data were normalized to the EV numbers. **M** Correlation between macrophage polarization obtained from FCM results and relative metabolic EVs-iNOS levels. The shaded red area denotes the 95% confidence band. $R^2 = 0.9732$. **N** Schematic illustration of iNOS knockout in RAW 264.7 cells, and corresponding ACTIVITY readout of metabolic EVs-iNOS levels in these transfected cells with different phenotypes ($n = 3$ independent experiments). This metabolic EVs-iNOS levels in transfected macrophage were normalized to that of wild-type RAW 264.7 pro-inflammatory macrophage. Photo credit: Ru-Jia Yu. **O** Schematic illustration of the collection and polarization of mouse primary macrophage and corresponding ACTIVITY readout of metabolic EVs-iNOS levels ($n = 3$ independent experiments). Photo credit: Ru-Jia Yu. Data are presented as mean ± SD, unless otherwise stated. Source data are provided as a Source Data file.

a home-designed miniaturized device, which enables rapid multi-channel readout for the metabolic activity of EVs-iNOS (see supporting information for detailed design and prototype specification, Supplementary Fig. 16), showing comparable performance with a commercial electrochemical workstation (Supplementary Fig. 17). Given the consistent elevated metabolic EVs-iNOS levels were consistently observed in pro-inflammatory phenotypes, including both RAW 264.7 cells and mouse primary macrophages, we propose that this metabolic activity

of EVs can be universally used to describe macrophage polarization, offering an alternative cell-free biomarker for internal organ inflammation.

## Application of ACTIVITY strategy for pneumonia diagnostics
To further promote this ACTIVITY method for point-of-care diagnosis of human inflammatory-related diseases, primary human monocyte-derived macrophages were used to validate the metabolic EVs-iNOS

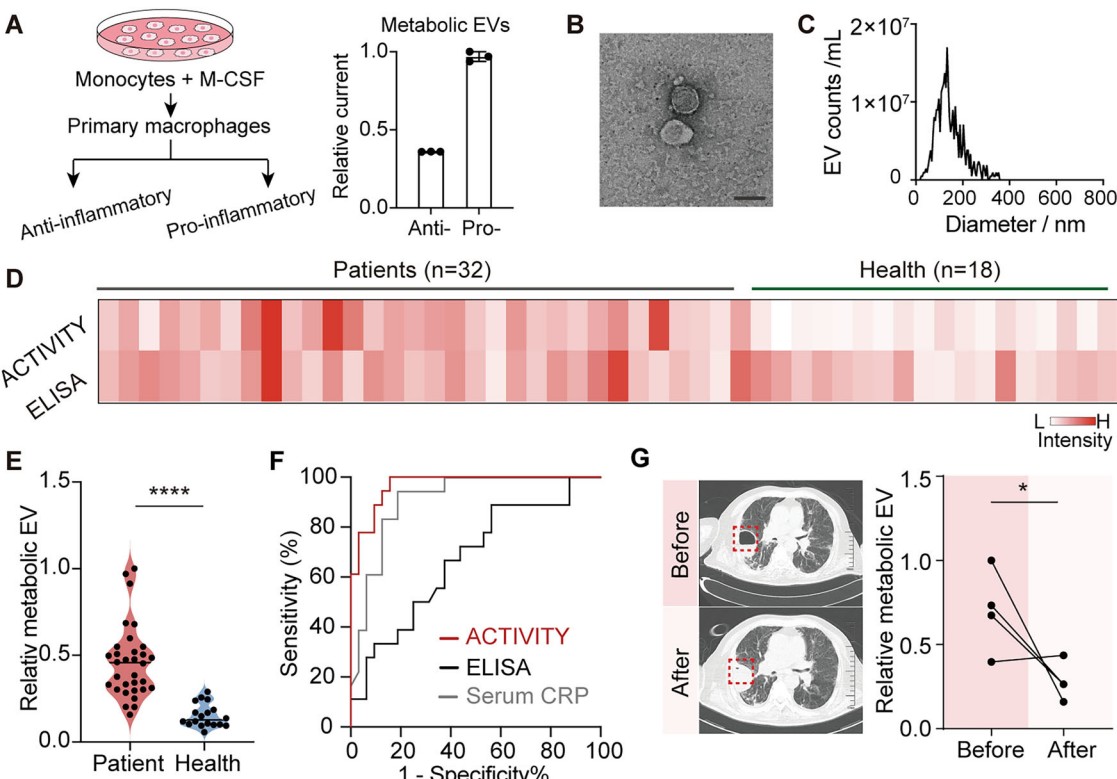

**Fig. 4 | Clinical application of the ACTIVITY strategy for pneumonia diagnostics. A** Schematic illustration of the differentiation and polarization of human SC monocytes, and corresponding ACTIVITY readout of metabolic EVs-iNOS levels (*n* = 3 independent experiments). Photo credit: Wei-Yi Ma. Data are presented as mean ± SD. Representative TEM image (**B**) and particle size distribution (**C**) of EVs isolated from BALF of a pneumonia patient. The experiment was independently repeated three times with similar results. **D** The ACTIVITY (top) and ELISA (bottom) readouts of clinical samples, including 32 pneumonia patients and 18 healthy individuals. The signal intensities were averaged over triplicate measurements of each sample and normalized by min-max normalization. **E** Statistical analysis of the metabolic EV-iNOS levels in pneumonia patients and healthy controls. **F** ROC curves for the metabolic EV-based ACTIVITY method (red line), the iNOS-based ELISA method (black line) and the serum parameter CRP (gray line). **G** Left: Representative CT images of pneumonia patient before and after clinical treatment. Right: Evaluation of the metabolic EV-iNOS levels in BALF samples from pneumonia patients (*n* = 4) before and after treatment. Both the ACTIVITY and ELISA data were normalized to the EV numbers. Statistical comparisons were made using repeated-measures two-way ANOVA with Sidak's multiple comparisons test (**E**, **G**). Source data are provided as a Source Data file.

detection. A non-cancerous human monocyte model, SC cells were differentiated into primary human macrophages and subsequently polarized into pro-inflammatory and anti-inflammatory phenotypes by culturing with differentiation and polarization medium[29]. The expression of CD86 was employed for the characterization of pro-inflammatory polarization (Supplementary Fig. 18). As shown in Fig. 4A, metabolic EVs-iNOS levels in these SC pro-inflammatory macrophages clearly distinguished from those of in SC anti-inflammatory cells. To demonstrate the potential clinical practice, we applied this ACTIVITY assay for the metabolic EVs-iNOS measurement in BALF samples and explored its promise as a pneumonia-related biomarker. BALF is widely recognized to provide a direct window for rapid lung disease evaluation, containing abundant EVs primarily derived from lung macrophages[30]. TEM characterization (Fig. 4B) and size distribution analysis (Fig. 4C) confirmed the abundant presence of EVs in BALF samples. Western blot analysis further verified the presence of iNOS in BALF-EVs collected from a pneumonia patient (Supplementary Fig. 19). Although the morphology of EVs isolated from patient was comparable to that of healthy donor (Supplementary Fig. 20), a slightly higher concentration was observed in the patient sample. Thus, we deduce that the metabolic profiling of EVs-iNOS in this fluid would possibly be used for the diagnosis of lung inflammation. BALF samples were first centrifuged to remove impurities, and the resulting EVs were subsequently collected, lysed, and subjected to ACTIVITY assaying to determine the metabolic EVs-iNOS levels. We first analyzed EVs from BALF samples collected from 32 pneumonia patients and 18 healthy

controls, with patient groups defined according to the integrated clinical diagnosis (see supplementary information for clinical information for these participants). Figure 4D shows the profiling results of these clinical samples, which reflect the intrinsic EV-associated activity rather than variation in EV numbers, as all the EVs samples were pre-quantified before analysis. The normalized metabolic EVs-iNOS levels were significantly higher in the patient samples compared to the healthy controls. Scatter plots and significance analysis were performed using the relative current signals, demonstrating an excellent discrimination between healthy individuals and pneumonia patients (Fig. 4E). For the same BALF samples, we conducted a comparative analysis using the conventional ELISA method for iNOS protein detection in EVs. Receiver operator characteristic (ROC) curve analysis was utilized to evaluate the accuracy of these two methods. As a comparison, a representative indicator of serum C-reactive protein (CRP) level was supplemented in Fig. 4F, showing the area under the curve (AUC) of 0.91. The ACTIVITY method for metabolic EVs-iNOS exhibited an AUC of 0.97, showing better performance than ELISA with an AUC of 0.67 (Fig. 4F). It demonstrated that the iNOS protein level in BALF-EVs was less representative of the metabolic-related inflammatory profiling compared with the ACTIVITY assay. This discrepancy between total iNOS protein level and its biological activity is probably due to the heterogeneity of iNOS molecules, as only a fraction exists in enzymatically active dimers, while monomers or partially folded proteins contribute to the total protein detected by ELISA[31]. In contrast, this proposed ACTIVITY method mainly relies on the enzymatic

activity of EV unit, which not only provides a stronger correlation with the inflammation response in the organ, but also offers greater sensitivity enabling precise detections even at low EV concentrations. Across all clinical samples tested, the ACTIVITY assay for detecting EVs-iNOS could better reveal the inflammation in the lung. Furthermore, this ACTIVITY method was used to assess the clinical treatment of pneumonia patients. An independent cohort of 4 patients was selected, with their BALF samples collected before and after clinical treatment. As demonstrated in Fig. 4G, most of the patients exhibited a decreased metabolic EV-iNOS level following clinical treatment, suggesting that this ACTIVITY method could effectively be used for monitoring therapeutic effects (Supplementary Fig. 21, 22). Therefore, as a radiograph-free alternative, this cascade amplification for metabolic EVs-iNOS detection facilitates better clinical diagnosis and monitoring of lung inflammation. It is worth noting that iNOS expression may be elevated in neutrophils or other immune cells during inflammation, and BALF samples contain part of these cell-derived EVs. However, comparative analysis of EVs from neutrophils, macrophages, and endothelial cells, which represent the major EV sources, showed that macrophage-derived EVs exhibited higher EV-iNOS activity (Supplementary Fig. 23). This may be attributed to the high functional plasticity of macrophages and the enhanced cellular crosstalk mediated by EVs that occurs throughout the inflammatory process. Further studies are needed into the precise mechanisms by which macrophages deliver metabolic markers via EVs to strengthen these findings. Meanwhile, there are some limitations in this study. We mainly focus on the bulk EV analysis, which may overlook the heterogeneity of EV subpopulations presented in biofluids, including both inter- and intra-subpopulation heterogeneity[32]. First, immunobeads mediated pull down-method could be employed to isolate EVs from specific cell types, which highly depends on the specificity and affinity of antibodies. Second, density-gradient centrifugation and size-exclusion chromatography can improve EV purity and enable the separation of EVs with different size distributions, while sacrificing the EV production in each subtype, thereby requiring larger sample volumes or more sensitive detection methods. With technological advances, integration of electrochemical platform with single entity approaches could enable subtype-specific EV profiling, which may offer higher sensitivity and more detailed insights into metabolic activity at the single vesicle level. In addition, macrophages exhibit functional plasticity in the lung, and their responses to the inflammatory microenvironment are dependent on the specific pulmonary disease context. Future single-vesicle analysis that accesses the heterogeneity of EV metabolic signatures may therefore provide fundamental insights into the association between EV function and the detailed disease progression.

In summary, we found that the bioactivity of macrophage-derived EVs serves as an independent metabolic biomarker for macrophage phenotype profiling and, importantly, for inflammation detection in the lung. A cascade signal amplification strategy was developed for the detection of iNOS activity of EVs, which leveraged the biocatalysis of EVs and the electrocatalysis of defective $WS_2$ QDs. The proposed ACTIVITY strategy was used for EV quantification, showing nearly three orders of magnitude higher sensitivity than that of conventional protein-based ELISA method. We further explored its clinical application for detecting lung inflammation using BALF samples from patients and healthy controls, with this ACTIVITY approach achieving an accuracy of over 96%. Due to the high indicative features of metabolically active EVs, this method facilitates rapid evaluation of pneumonia treatment in a non-radiative manner. Though we have demonstrated the advantages of this ACTIVITY strategy for inflammation detection, given that metabolic features are associated with many diseases, this method of directly assaying metabolically active EVs could be used for broader clinical applications. Further technical developments, such as combining portable signal readout instrumentation and magnetic

separation to facilitate EV collection, could promote the method as a versatile detection platform for disease profiling.

## Methods

### Ethical statement
The animal experiment was performed following the ethical guidelines and approved by the Committee for Animal Research of Nanjing University of Posts and Telecommunications (approval no. 2022012). The BALF samples from pneumonia patients and health controls were collected by the First Affiliated Hospital of Soochow University. All research participants provided written informed consent. The ethical approval of the study was obtained from the Ethics Committee of the First Affiliated Hospital of Soochow University (approval no. 2023455).

### Chemicals and reagents
All chemicals and reagents were used as received without any purification. $W_2O_3$, $Na_2S$, HCl, $NaNO_2$, glutathione, and BLZ945 were purchased from Aladdin reagent company (Shanghai, China). NADPH, L-arginine, lipopolysaccharide, RIPA lysis buffer and NP-40 lysis buffer were purchased from Beyotime Biotechnology (Jiangsu, China). Antibodies, including APC anti-CD86, PE anti-CD206, anti-CD63, and anti-iNOS, were purchased from Abcam (Cambridge, MA, USA). Anti-CD9 and anti-Tsg101 were purchased from Huabio (Zhejiang, China). Mouse and human iNOS ELISA kits were purchased from Beijing 4 A Biotech (Beijing, China) and Ruixin Biotech (Quanzhou, China), respectively.

### Preparation of defective $WS_2$ QDs
A bottom-up method was used for the preparation of $WS_2$ QDs. Briefly, 2.31 g $W_2O_3$ powder was dissolved in an aqueous NaOH solution with a concentration of 0.5 mM at pH 11. The solution was sonicated until fully dissolved, showing as transparent and colorless. Next, 1 mL of the $W_2O_3$ solution was mixed with 10 mL GSH solution (10 mg/mL) under vigorous stirring, followed by the addition of 0.5 mL 1 mM $Na_2S$ solution. After stirring for an additional 10 min, 1 M HCl was added dropwise to adjust the pH to 6 ~ 7. The solution changed from colorless to a clear green color, indicating the formation of $WS_2$ QDs. For controllable defect engineering, $H_2O_2$ solutions of different concentrations were further introduced to the QDs solution. After staying for 5 min in the dark, the mixed solution was heated at 60 °C for 30 min.

### Preparation of NO solution
The gaseous NO was prepared by an acid-base reaction between sulfuric acid and sodium nitrite. Briefly, by dropwise adding 4 M $H_2SO_4$ into a deoxygenated 2 M nitrite sodium solution, NO gas was produced immediately and then was double perfused through 1 M NaOH solution for purification. NO solution at a concentration of 1.8 mM was further prepared by continuously bubbling NO gas into PBS solution for 30 min to get saturated at 20 °C. Degassed PBS solution was used to dilute the saturated NO solution to prepare NO solutions at different concentrations for electrochemical measurement.

### Electrochemical measurements for NO
Electrochemical measurements were conducted on a CHI 660E electrochemical workstation with a three-electrode system with glassy carbon, Pt, and Ag/AgCl electrodes serving as working, counter, and reference electrodes, respectively. The glassy carbon electrode was used to prepare the QDs-modified electrode. Before modification, the electrode was first polished with 0.05 μm alumina slurry on a polishing cloth and then sonicated successively in ethanol and deionized water for thorough cleaning. The QDs-modified electrode was prepared by coating with a mixture of QDs and Nafion solution. Briefly, 8 mg defective $WS_2$ QDs were dispersed into 1 mL water/ethanol solution (4:1, v/v), together with 80 μL 5 wt% Nafion solution. This mixture was then sonicated for >30 min to form homogeneous dispersion. 6 μL of this suspension was dropped on the cleaned glassy carbon electrode

and dried at room temperature to form the QDs modified electrode for electrochemical detection of NO. 1xPBS at pH 7.4 was used as an electrolyte for all the electrochemical measurements. The electrochemical experiments were all conducted in a Faraday cage.

## RAW 264.7 cell culture and macrophage polarization

A mouse macrophage cell line, RAW 264.7 (TIB-71, American Type Culture Collection) was employed as an in vitro model for macrophage polarization. Cells were cultured in Dulbecco's modified Eagle's medium supplemented with 10% fetal bovine serum and 1% penicillin-streptomycin, maintained in a humidified environment with 5% $CO_2$ at a 37 °C incubator. For the polarization, RAW 264.7 cells were first differentiated into anti-inflammatory phenotype by incubating with 20 ng/mL Interleukin-4 (IL-4) for 24 h. After washing with PBS to remove the remaining chemicals, the macrophages were polarized toward pro-inflammatory phenotype by incubating with 100 ng/mL lipopolysaccharide (LPS) for 4 h, 8 h, 12 h, and 24 h. Macrophages treated with 5 µg/mL BLZ945 were respectively performed with the same experimental procedures to serve as a drug monitoring model. The macrophage polarization was characterized by the percentage of the expression level between CD86 and CD206.

## Gene construction and cell transfection

The pSpCas9(BB)-2A-EGFP (PX458) plasmids (see details in supplementary Table 1) carrying mNos2-sgRNA was purchased from Abologen (Chengdu, China). RAW264.7 cells in T25 dish were grown to ~70% confluency and then washed with PBS buffer. According to the manufacturer's instruction, cell transfections were performed by using Lipofectamine 3000 Reagent (Invitrogen, USA) at a final plasmid concentration of 2 µg/mL. After 8 h of incubation, cells were washed with PBS and then cultured for additional 24 h before performing experiments. GFP expression was observed clearly, which confirms the successful transfection and the knockout of iNOS.

## Collection of primary mouse peritoneal macrophages

Balb/c mice (female, 6–8 weeks, purchased from the Nanjing Qinglong Mountain Animal Farm) were used for the isolation of primary peritoneal macrophages. All mice were housed at room temperature (18–24 °C) with relative humidity (40-60%) and a 12 h day-night cycle with consistent food and water. 6 mice were used for peritoneal lavage harvesting after being killed by cervical dislocation. DMEM medium was injected into the peritoneal cavity using a sterile syringe. After gently massaged to detach resident peritoneal cells, the peritoneal lavage fluid was then carefully aspirated and collected into sterile tubes. The collected cells were centrifuged and resuspended in cell culture medium. After 4 h incubation, non-adherent cells were removed by washing and the adherent cells were considered primary peritoneal macrophages and used for subsequent experiments.

## Monocytes cells differentiation and polarization

Human peripheral blood monocytes, SC cells (CBP61294, Cobioer, Nanjing, China) were cultured in IMDM supplemented with 10% FBS and 0.05 mM 2-mercaptoethanol, 0.1 mM hypoxanthine and 0.016 mM thymidine. SC cells were differentiated into primary human macrophages by treatment with 15 ng/mL M-CSF1 for 6 days, with half the medium replaced on the third day. The differentiated cells were further treated for 24 h with 20 ng/mL IL-4 and 20 ng/mL IL-13 to polarize into anti-inflammatory, whereas 100 ng/mL LPS and 20 ng/mL IFN-γ to get pro-inflammatory macrophages, respectively.

## Flow cytometry for determination of macrophage polarization

The macrophage polarization was determined by flow cytometry using antibodies against CD86 and CD206. Specifically, RAW 264.7 cells differentiated into anti-inflammatory phenotype were seeded in a 6-well plate at $8 \times 10^4$ cells/ well and cultured in the incubator for 24 h.

After treatment with LPS or BLZ945, cell suspension was replaced by freshly prepared medium containing 1 µg/mL anti-mouse CD86 (labeled with APC) and anti-mouse CD206 (labeled with PE). The macrophages were incubated for another 30 min at 37 °C and carefully washed with PBS. Fluorescence signals from the labeled cells were obtained by flow cytometry and FlowJo software was used for analysis. For the human macrophages differentiated from SC cells, APC-anti human CD86 (Biolegend, USA) was used to characterize the pro-inflammatory phenotype.

## EV collection, characterization and analysis

EVs were collected from the cell culture supernatants or bronchoalveolar lavage fluid. The ethical approval of the study was obtained from the Ethics Committee of the First Affiliated Hospital of Soochow University (2023455). The detail information of cohorts involved in this experiment were provided in the Supplementary Table 2. The case for RAW 264.7 cell-derived EVs, vesicle collection was performed after culturing cells with a vesicle-depleted medium (containing dialyzed fetal bovine serum) for 48 h. EVs were enriched from the cell culture supernatants by ultracentrifugation. Briefly, the samples were first centrifuged at $500 \times g$ for 5 min and $10,000 \times g$ for 90 min to remove cells and large debris, respectively. After flited through a 0.22 µm Millipore filter, the supernatant was then ultracentrifuged at $100,000 \times g$ for 2 h to pellet EVs. The obtained EVs samples were resuspended in PBS, aliquoted and stored at −80 °C for further usage.

Morphological characterization of EVs was performed by transmission electron microscopy (TEM) with 3% uranyl acetate staining. Nanoparticle tracking analysis (NTA) was performed to demonstrate the size distributions and concentrations. After the lysis of EVs, Western blot analysis was conducted for protein characterization according to the manufacturer's protocols. For EVs collected from both cell culture supernatants and BALF samples, TEM, NTA, and Western blot analyses were performed to confirm the EV characteristics.

In the other case, for EV collections from bronchoalveolar lavage fluid, exosomes extraction kit (Beijing Baiao Laibo Technology Co. Ltd, China) was employed after centrifugation at $8000 \times g$ for 5 min to remove sputum and other impurities. The collected EVs samples were further lysed for ELISA and electrochemical analysis. EV protein content was firstly quantified using a BCA protein assay (Beyotime Biotech, China), after which iNOS ELISA and the ACTIVITY analysis were performed using equal EV input. This normalization allowed us to compare the intrinsic iNOS activity per EV rather than the total EV yield. For the BCA assay, EVs were lysed with RIPA lysis buffer to ensure complete protein extraction. In contrast, for iNOS ELISA and enzymatic activity analysis, EVs were lysed with a freshly prepared NP-40 buffer supplemented with 1 mM PMSF, which preserves protein conformation and enzymatic activity. For the ACTIVITY measurement, specifically, 200 µL of lysis buffer was added to the EV precipitate, followed by incubation with 20 µL of 5 mM L-Arg and 20 µL of 1 mM NADPH. Then this mixture containing NO, the product of the enzymatic reaction, was transferred to 1.8 mL PBS solution in an electrochemical cell. The i-t curves were subsequently recorded for electrochemical measurements. The electrochemical readouts for metabolic EVs-iNOS were obtained after background subtraction, which accounted for the possible interference of lysis buffer and enzyme substrate in PBS. A min-max normalization was subsequently used unless otherwise stated. Regarding the detection time, EV lysis requires ~15 min, the enzymatic reaction takes ~30 min, and the electrochemical readout could be obtained within a few minutes.

## ELISA

iNOS concentrations in vesicles were determined by commercial ELISA assay kits according to the manufacturer's instructions. Mouse iNOS ELISA kit (Beijing 4 A Biotech, Beijing, China) was used for the detection of RAW 264.7 cell-derived EVs; while human iNOS ELISA kit (Ruixin

Biotech, Quanzhou, China) was used for the BALF collected EVs. Generally, a 96-well plate was pre-anchored with capture antibody and blocked with bovine serum albumin. EV lysis with different dilutions was added to the plate and incubated for 90 min. After twice washing with PBST solution, biotinylated detection antibodies were employed for 1 h's incubation. Following successive incubation with streptavidin-conjugated horse radish peroxidase, enzymatic substrates and stop solution, the absorbance at 450 nm was measured with a microplate reader.

## Statistics and reproducibility

At least three independent experiments of each type have been conducted and have produced consistent results. All the results are presented as mean ± standard deviation (SD) unless otherwise stated. The statistical significance was analyzed by Graph Pad Prism 10 software using a two-way repeated measures ANOVA with Sidak's multiple comparisons test.$*p < 0.05$, $**p < 0.01$, $***p < 0.001$ and $****p < 0.0001$ were considered as statistically significant differences. No statistical method was used to predetermine sample size. No data were excluded from the analyses. The experiments were not randomized. The investigators were not blinded to allocation during experiments and outcome assessment.

## Reporting summary

Further information on research design is available in the Nature Portfolio Reporting Summary linked to this article.

## Data availability

Source data are provided as a Source Data file. Source data are provided with this paper.

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

## Acknowledgements

This research was supported by National Natural Science Foundation of China (22577056, X.G. D., 22276089, R.J.Y. and 62288102, L.H.W.) and Basic Research Program of Jiangsu (BK20243057, X.G. D. and BK20253006, L.H.W.). We thank P.C. Zhou (Affiliated Hospital of Nantong University) for the assistance of western blot analysis.

## Author contributions

R.J.Y. and X.G.D. designed the research. H.Y.X., W.Y. M., Y.W.Z., K.L.W., and W.B.G. conducted the main experiment and data analysis. H.Y.X., Y.W.Z., Z.F.Y., K.R.X. and X.W. contributed to the project investigation. W.Y. M., Y.W.Z., H.J.Z. prepared the figures. L.H.W. and X.G.D. supervised the project and provided suggestions for the research. R.J.Y.,

W.Y.M., Y.W.Z., and X.G.D. wrote the paper. All authors discussed the results and reviewed the manuscript.

## Competing interests

The authors declare no competing interests.
