## [Transparent Peer Review file · Nature Communications]

Amplifying metabolic profiling of extracellular vesicle dynamics with ACTIVITY

Corresponding Author: Dr Xianguang Ding

Version 0:

Reviewer comments:

Reviewer #1

(Remarks to the Author)

In this manuscript, the authors reported a method that combines the biocatalysis of EV iNOS activity with the electrocatalysis of defective tungsten disulfide quantum dots (WS₂ QDs) to detect iNOS activity in EVs. They claimed it might be a marker for characterizing macrophage polarization and inflammatory disease. However, the current results can not support their conclusion, and many major issues need to be addressed.

1. Mounting evidence has indicated that ultracentrifugation alone can co-isolate many unknown contaminants (including proteins) during isolating EVs from complicated biological samples and thus induce false signals in the following assays. Therefore, this study must first demonstrate the purity of the EV preparations and that iNOS proteins are indeed within EVs but not the contaminants using more strict EV isolation methods, such as density gradient centrifugation.
2. As a potent biomarker, it is vital to define the distribution pattern of functional iNOS proteins, are there in all EV subtypes or only certain EV subtypes? The absolute concentrations of iNOS protein per EV, as well as its alteration in different conditions, should be assayed.
3. Also, the exact location of the iNOS protein in EVs, such as surface or inside, should be assessed.
4. The authors claimed that "this protein (iNOS) concentration is less representative for the metabolic-related inflammatory profiling". If so, what is the exact reason for the poor correlation between protein concentrations and bioactivities? If some iNOS proteins carried by EVs are functionally compromised?
5. To prove the specificity of this assay, it is needed to modulate EV iNOS expression levels by overexpression or knockout of iNOS in parent cells and then perform EV isolation and detection.
6. The protocol of M1-like macrophage (M1) induction used in this study is not standard, and mouse macrophage cell lines (RAW 264.7 cells) were pretreated with IL-4 and then stimulated with LPS or BLZ945. In fact, LPS alone is sufficient to induce M1 macrophages, please explain the reason. Also, BLZ945 is a CSF-1R inhibitor, is there any evidence to support that it can induce M1 macrophages and what is the underlying mechanism to do so? The phenotype of macrophages used in this study should be carefully assessed, as they are unlikely classic M1 macrophages.
7. Cell lines may have varied biological properties compared to the primary cells, particularly in immune cells, and thus primary macrophages should be included in each test.
8. BALF samples contain diverse cell types, but the authors claimed that "BALF containing abundant EVs primarily derived from lung macrophages". To support this statement, it is required to assess the exact percentages of the lung macrophage-derived EVs in the total BALF EVs.
9. In a clinical translation aspect, the sample number for patients of each group is too small, and these findings need to be verified in a more large cohort of patients.
10. Many important information in the methods section was missing. For example, the lysis buffer of EVs for electrochemical measurements is unclear. It seems that RIPA lysis buffer was used, however, RIPA buffer contains high concentrations of detergents, and whether they can affect the activities of the detected proteins should be determined. And thus more details must be added in the methods sections that allow the replication of this study.

Reviewer #2

(Remarks to the Author)

In the current manuscript, Yu et al. demonstrate that the metabolic-related inducible nitric oxide synthase (iNOS) activity of macrophage-derived EVs serves as an effective biomarker for phenotypic profiling and further evaluating lung inflammation. The authors developed an ACTIVITY method based on extracellular vesicles (EVs) diagnosis, utilizing the functional

metabolic activity of iNOS in macrophage-derived EVs rather than static molecular cargo. Overall, the study is well-designed and interesting. Somehow, the study looks preliminary and there are many important concerns that need to be addressed.

Some suggestions/questions:

1. The 14-day stability test (Fig. 3F) solely assesses storage stability. The study overlooks mentioning the stability of defects in WS₂-DH QDs following prolonged storage or repeated uses (like performance alterations after multiple CV tests). It is advisable to present pertinent data to bolster the practical application feasibility.
2. Potential Contamination Concerns in EV Preparations: The ultracentrifugation protocol (100,000g for 2h) may inadvertently co-isolate apoptotic bodies or protein aggregates, particularly in BALF samples. Data pertaining to EV-specific markers (e.g., CD63, TSG101) are absent (Fig. 3B, S5).
3. For experiments involving iNOS in this study, it is advisable to employ NOS2 gene knockout mice as controls to affirm the study's accuracy.
4. The study presumes iNOS activity remains constant throughout EV storage (-80°C) and lysis. No data are provided regarding time- or temperature-dependent activity loss.
5. LPS was utilized to stimulate M1-type macrophage polarization in this study, whereas the conventional method for inducing M1-type macrophage polarization involves LPS combined with IFN- γ . The authors should further refine their research protocol.
6. Currently, only the RAW 264.7 cell line is used. Incorporating experimental results from primary macrophages (like human peripheral blood mononuclear cell-derived macrophages or BMDM) would bolster the conclusions' universality.
7. The patient group (n=18) and control group (n=11) sample sizes are relatively modest, and pneumonia subtypes (bacterial/viral/non-infectious) or severity stratification are unspecified. Expanding the sample size and performing stratified analysis would validate the method's broad applicability.
8. Imaging diagnosis serves as the gold standard in the article, yet it lacks comparison with existing inflammatory markers (such as serum CRP, PCT). Correlation analysis should be incorporated to elucidate its additional diagnostic merit.
9. Numerous studies indicate that iNOS can be produced by various immune cells, including neutrophils, dendritic cells, and MDSCs. How this approach mitigates this clinical application limitation remains to be addressed.

Reviewer #3

(Remarks to the Author)

Novelty- less studies have been performed on metabolic profiles of EVs compared to proteomic or genomic analysis of EVs. This is correct. The arginine based "activity probe" appears to detect iNOS metabolic activity with sensitivity when incubated with the EVs. Personally i dont know of a paper where an "ACTIVITY" probe was used to determine NOS activity in EVs. From that perspective the work has potential significance.

General concerns: Some of the writing and comments throughout need rewriting

Inflammation is referred to very generally throughout, given the complexity of the macrophage EV driven inflammation which differs across each distinct lung disease i'd be cautious of general marks also in terms of polarisation. M1 like macrophage cytokines have been associated with pneumonia but the idea of pro-inflammatory M1 macrophages and M2 macrophages suppressing inflammation is likely a simplification as macrophages in lung exhibit plasticity in response to different stimuli.

Specific examples

... from L-arginine to generate line 60 missing word end of sentence

'In the inflammation in internal organs such as lung'.... Line 60 badly phrased

Approach for evaluating internal organ inflammation- general remark -line 68

Figure 1: schematic looks ok except NOS blot not very convincing, many bands on it

Figure 2: Design and evaluation of ACTIVITY probes- no comment as not my area

Figure 3: Why are the authors using RAW 264.7 cells from mice to see if EV metabolic signature characterizes macrophage polarization? That would make sense if they were going to validate approach in a mouse after but human BALF samples are used for further validation. Is there not a more relevant model, alveolar macrophage, even macrophages from derived human PBMCs (although blood and airway macrophages differ)

Figure 4: Clinical details: Are there details re the 18 pneumonia BALF samples and patient characteristics? What state of disease progression etc?.

Writing: Re the sentence 'BALF is widely recognized as an ideal less-invasive fluid for rapid disease evaluation....' BALF is not widely regarded as less invasive as it is obtained via bronchoscopy which is invasive, i'd rephrase. BALF samples are often obtained via a biobanking program, is this the case here?

Figure 4 part B-TEM of BALF EVs could be better- 4B, is it from healthy or pneumonia donors? Normally both shown.

Part C- assume this is by NTA, id state it is a representative of a graph from 1 healthy or 1 pneumonia patient?

Further comments, the authors have shown there is more metabolic NOS EV activity in BALF from pneumonia patients which might be promising with further characterisation.

In addition to clinical info what other characteristics do these EVs have, are there generally more EVs by NTA in the pneumonia patients for example?

How does this functionally relate back to macrophages, BALF contains EVs of many cell types including EVs from epithelial cells, neutrophils etc which may also can have elevated NOS activity. If macrophages were isolated from BALF and their ACTIVITY measured this would be more convincing

Reviewer #4

(Remarks to the Author)

The authors have developed a novel methodology for the rapid profiling of EV-iNOS activity, which they have shown to be more sensitive than existing ELISA technologies for measuring EV metabolic activity. Using the developed assay, they quantify iNOS activity in macrophage-derived EVs, which is reflective of the metabolic state of the macrophage and its inflammatory state. They then successfully show differences in macrophage-EV metabolic activity in individuals with pneumonia and healthy counterparts.

Thus, the authors provide promising evidence for the use of EV-iNOS activity as a potential biomarker for lung inflammation, using this novel EV assay. The paper is well-written and concise, but more detail is needed in the EV methodology sections for readers to be able to replicate the results and to meet MISEV standards.

Major corrections

- Lines 340-346 – It is not clear in the methods whether EV characterisation was performed for BALF-derived EVs. Please indicate which technologies were performed to confirm the presence and characterisation of EVs from the BALF samples.
- General – There is evidence of TEM and NTA profiling of EVs, but according to MISEV 2023 guidelines, EV tetraspanin measurements are recommended when reporting EV data. Unless there is a convincing argument against this, please present data for CD9/CD63/CD81 expression on the EVs of interest.
- The term 'rapid assay' is used throughout, but it is not clear how long the assay actually takes. Please state in the methods or results the approximate time for the assay.
- Is it a "non-invasive" method for pneumonia treatment evaluation if it requires BALF collection? This needs re-considering.

Minor corrections

- Line 45 – remove 'etc'.
- Line 308 - 'in vitro' should be italicised throughout.
- Line 330 - State how many BALF samples were used in the study.
- Lines 349-350 - State the manufacturer for the ELISA kit and details for the capture antibody.

Lucy Fairclough and Georgina Hopkins

Reviewer #5

(Remarks to the Author)

The manuscript presents an electrochemical assay for measuring the activity of inducible nitric oxide synthase (iNOS) available on extra cellular vesicles by analyzing its enzymatic product, nitric acid. The manuscript is interesting and well written and can be published after addressing the following comments.

1. The clinical sensitivity and specificity of the assay should be determined and quoted against the primary method used for diagnosing pneumonia and not just ELISA. The primary diagnostic method should be discussed.
2. The advantage of using WS2 to measure the sensor response over other carbon-based or catalytic metallic electrodes should be demonstrated using experiments.
3. The normalization used in the different figures is not clear to me, for example, it is written that the CD86/CD206 is used in Figure 3G but it is not clear how other signals have been normalized.
 - a. What is the justification for using CD86/CD206?
 - b. How are the electrochemical signals normalized?
 - c. How are each one of the signals normalized, this should be included in the figure captions and justified in the manuscript

Reviewer #6

(Remarks to the Author)

Reviewer #7

(Remarks to the Author)

Version 1:

Reviewer comments:

Reviewer #1

(Remarks to the Author)

In this revision, the authors have answered some of my comments; however, there are some major issues that need to be addressed.

1. In cellular iNOS knockout assay, besides the GFP expression, more direct results (iNOS protein expression) should be provided using western blotting. Also, the effect of cellular iNOS overexpression on EV properties is needed.

2. The concerns about the induction method of macrophages used in this study still exist. If the cell models cannot correctly represent the changes in diseased state, it may result in misleading or false conclusions.

3. The authors claimed that "IL-4 pretreatment was applied to establish an initial M2-like anti-inflammatory state, thereby providing a defined baseline from which dynamic phenotypic changes could be examined following inflammatory stimulation". However, the in vivo IL-4 levels in physiologic state are relatively low, and previous papers have reported that the IL-4-to-LPS induction can induce non-canonical proinflammatory M2INF macrophages (e.g., PMID: 37149865) that display a different phenotype compared to M1.

4. The authors also claimed that "BLZ945 was used as a phenotype-modulating perturbation based on prior evidence that inhibition of CSF-1R signaling by BLZ945 suppresses M2 polarization and shifts macrophage phenotypes toward a more pro-inflammatory state in vivo. However, in this study, the in vitro culture medium does not contain its ligand (CSF-1), so what's the exact role of the CSF-1R inhibitor?"

5. Because of the biological difference between the cell lines (RAW 264.7 cells) and the primary cells, the key finding of this study (including above) should be validated using primary mouse or human macrophages.

Reviewer #3

(Remarks to the Author)

I am content the authors addressed the issues that I raised in the original review, both in the text and experimentally. I believe manuscript is now to be fit for publication.

Reviewer #4

(Remarks to the Author)

The resubmitted manuscript has made all required changes and is now suitable for publication.

Reviewer #5

(Remarks to the Author)

The authors have thoroughly addressed my comments. I have not further comments.

Reviewer #6

(Remarks to the Author)

Reviewer #7

(Remarks to the Author)

Version 2:

Reviewer comments:

Reviewer #1

(Remarks to the Author)

The authors have addressed my questions, and it is recommended for publication in this journal.

Reviewer #1:

In this manuscript, the authors reported a method that combines the biocatalysis of EV iNOS activity with the electrocatalysis of defective tungsten disulfide quantum dots (WS₂ QDs) to detect iNOS activity in EVs. They claimed it might be a marker for characterizing macrophage polarization and inflammatory disease. However, the current results cannot support their conclusion, and many major issues need to be addressed.

RESPONSE: We sincerely appreciate the reviewer's careful evaluation of our manuscript and the constructive suggestions regarding the strength of our conclusions. Please find the details below.

Q1. Mounting evidence has indicated that ultracentrifugation alone can co-isolate many unknown contaminants (including proteins) during isolating EVs from complicated biological samples and thus induce false signals in the following assays. Therefore, this study must first demonstrate the purity of the EV preparations and that iNOS proteins are indeed within EVs but not the contaminants using more strict EV isolation methods, such as density gradient centrifugation.

R1. We thank the reviewer for pointing out this potential co-isolation issue during the ultracentrifugation-based EVs isolation. In response to the concern regarding potential co-isolation of protein aggregates, we have supplemented additional characterization experiments. First, we performed Western blot analysis to confirm the presence of characteristic EV markers, including CD9, CD63, and TSG101 (revised Figure 1C), supporting the successful enrichment of EVs.

Figure 1. (C) TEM image and Western blot analysis of macrophage-derived EVs. scale bar, 100 nm.

In addition, we further purified the obtained EVs samples using iodixanol density gradient ultracentrifugation (DGUC) to assess potential contamination following the MISEV2023 guidelines. The resulting particle fractions were quantified with resistive pulse sensing method and the encapsulated iNOS protein were quantified via ELISA method after a vesicle lysis process. Specifically, obvious four fractions of particles were observed, with F4 in the bottom of tube showing as protein aggregates. By normalizing the iNOS protein concentrations to particles numbers, layer 4 shows the minimum iNOS concentration. In other words, potential proteins contaminants are expected to contribute negligibly to iNOS sensing. Nonetheless, we sincerely appreciate the reviewer's reminder, and we will adopt more rigorous EV isolation protocols in our future studies to further ensure the specificity and reliability of our findings. We also add limitation discussions in the Main Text as follows:

“Meanwhile, there are some limitations in this study. Such as, we mainly focus on the bulk EV analysis, which may overlook the heterogeneity of EV subpopulations presented in biofluids, including both inter- and intra-subpopulation heterogeneity [32]. First, immunobeads mediated pull down-method could be employed to isolate EVs from specific cell types, which highly depends on the specificity and affinity of

antibodies. Second, density-gradient centrifugation and size-exclusion chromatography can improve EV purity and enable the separation of EVs with different size distributions, while sacrificing the EV production in each subtype, thereby requiring larger sample volumes or more sensitive detection methods. With technological advances, integration of electrochemical platform with single entity approaches could enable subtype-specific EV profiling, which may offer higher sensitivity and more detailed insights into metabolic activity at the single vesicle level. In addition, macrophages exhibit functional plasticity in the lung, and their responses to the inflammatory microenvironment are dependent on the specific pulmonary disease context. Future single-vesicle analysis that access the heterogeneity of EV metabolic signatures may therefore provide fundamental insights into the association between EV function and the detailed disease progression.”

Supplementary Figure (A) Standard curve for iNOS quantification using ELISA. (B) Four main particle fractions with F4 representing as protein aggregates. The number of particles in each fraction were obtained through resistive pulse sensing method and each iNOS concentrations were determined by ELISA. (C) iNOS concentration normalized to particle number for each fraction.

The Method Section has been detailed as follows:

“The case for RAW 264.7 cell-derived EVs, vesicle collection was performed after culturing cells with a vesicle-depleted medium (containing dialyzed fetal bovine serum) for 48 h. EVs were enriched from the cell culture supernatants by ultracentrifugation. Briefly, the samples were first centrifuged at 500 g for 5 min and 10000 g for 90 min to remove cells and large debris, respectively. After filtered through a 0.22 μm Millipore filter, the supernatant was then ultracentrifuged at 100,000g for 2 h to pellet EVs. The obtained EVs samples were resuspended in PBS, aliquoted and stored at -80 °C for further usage.”

Q2. As a potent biomarker, it is vital to define the distribution pattern of functional iNOS proteins, are there in all EV subtypes or only certain EV subtypes? The absolute concentrations of iNOS protein per EV, as well as its alteration in different conditions, should be assayed.

R2. Thank you so much for this constructive comment. EVs are indeed a heterogenous population, commonly classified into exosomes (30 ~ 150 nm) and macrovesicles (100 nm ~ 1 μm) and apoptotic bodies (> 1 μm), each with substantial variation in size and molecular composition. In our work, the EVs obtained by differential ultracentrifugation were predominantly below 150 nm, indicating that the collected

EVs were mainly exosome-enriched rather than microvesicles or apoptotic bodies. Also, from the DGUC results, iNOS were detected in multiple EV fractions (F1, F2 and F3), suggesting that iNOS is not restricted to a single EV subtype but is broadly present across macrophage derived EV populations. As can be calculated from the above Supplementary Figure c, the absolute concentration of iNOS was ~ 6 molecules/per EV in F1 and ~11 molecules/per EV in F2 and F3. For, the alterations in different conditions, such as polarization into different phenotypes, the concentration of iNOS shows clear difference as we demonstrated in our Main Text Figure 3. Importantly, all EVs were isolated, processed and quantified under identical protocols, allowing reliable comparison of iNOS content per EV across different groups and disease contexts. This highlights the potential of iNOS-carrying EVs as a quantitative and functional biomarker, with their concentration per vesicle and dynamic changes under distinct physiological or pathological states offering biologically relevant information.

We also appreciate the reviewer's constructive suggestions on this critical issue of EV subtypes. We have supplemented discussion in the revised manuscript and acknowledge that future studies incorporating single-EV or subtype-resolved analyses will further elucidate EV heterogeneity and may provide deeper mechanistic insights for disease diagnosis. The revised Main Text is as follows:

“Meanwhile, there are some limitations in this study. Such as, we mainly focus on the bulk EV analysis, which may overlook the heterogeneity of EV subpopulations presented in biofluids, including both inter- and intra-subpopulation heterogeneity [32]. First, immunobeads mediated pull down-method could be employed to isolate EVs from specific cell types, which highly depends on the specificity and affinity of antibodies. Second, density-gradient centrifugation and size-exclusion chromatography can improve EV purity and enable the separation of EVs with different size distributions, while sacrificing the EV production in each subtype, thereby requiring larger sample volumes or more sensitive detection methods. With technological advances, integration of electrochemical platform with single entity approaches could enable subtype-specific EV profiling, which may offer higher sensitivity and more detailed insights into metabolic activity at the single vesicle level. In addition, macrophages exhibit functional plasticity in the lung, and their responses to the inflammatory microenvironment are dependent on the specific pulmonary disease context. Future single-vesicle analysis that access the heterogeneity of EV metabolic signatures may therefore provide fundamental insights into the association between EV function and the detailed disease progression.”

Q3. Also, the exact location of the iNOS protein in EVs, such as surface or inside, should be assessed.

R3. Thanks for very much this constructive comment. Actually, determining the exact localization of protein molecules in EVs is critical for understanding both its biological function and its utility as a biomarker. To address this, we compared iNOS protein content and its biocatalytic activity in either lysed or intact EVs samples. If iNOS were exposed on the EV surface, protein levels and its catalyzed NO production would be similar regardless of the lysis process. However, in our experiments, ELISA measurements showed pretty low iNOS levels in intact EVs which is in the absence of lysis buffer, while the lysed EVs samples exhibited significantly higher protein content. Moreover, NO production gradually increased in the lysed EVs upon incubation with the enzyme substrates, whereas the intact EVs group produced minimal NO with no significant change over incubation time. These results indicate that the disruption of the intact EV

structure is essential for iNOS sensing, strongly suggesting that iNOS locates inside the EVs lumen rather than the surface. The revised Main Text is shown as follow:

“EVs derived from RAW 264.7 cells were isolated from the cell culture supernatant via the ultracentrifuge method. The isolated EVs showed a typical discoid morphology (Fig. 3A) with an average diameter of ~120 nm (nanoparticle tracking analysis in Fig. S7). Western blot analysis confirmed the presence of iNOS protein in these macrophage-derived EVs, together with characteristic EV transmembrane markers including CD9, CD63 and Tsg 101 (Fig. S8). **To explore the exact location of the iNOS protein within EVs, we quantified both iNOS protein content and its biocatalytic activity in (i) lysed and (ii) intact EVs. As shown in Fig. 3B, ELISA measurements revealed higher levels of iNOS protein in lysed EVs compared with intact EVs, indicating limited accessibility of iNOS without membrane disruption. Meanwhile, iNOS biocatalytic activity was assessed by measuring NO production using a NO fluorescent probe, diaminofluorescein-FM in the presence of enzyme substrates. Lysed EVs exhibited a clear, time-dependent increase in NO production, whereas intact EVs produce very little NO (Fig. 3C). These results demonstrate that the disruption of EVs structure is essential for iNOS sensing, strongly suggesting that iNOS is localized within the lumen of EVs rather than on the surface. For this reason, iNOS exhibited considerable stability during EV storage, with its biocatalytic activity remained nearly unchanged over 42 days at -80 °C (Fig. 3D). We believe this enhanced stability is attributed to the inside localization of iNOS, where lipid encapsulation provides effective protection during the long-term storage.**”

Figure 3. ACTIVITY for EV metabolic activity analysis. (B) Lysis-triggered protein detection. iNOS concentrations in (i) lysed and (ii) intact EVs samples determined by ELISA. (C) Normalized biocatalytic activity in (i) lysed and (ii) intact EVs.

Q4. The authors claimed that “this protein (iNOS) concentration is less representative for the metabolic-related inflammatory profiling”. If so, what is the exact reason for the poor correlation between protein concentrations and bioactivities? If some iNOS proteins carried by EVs are functionally compromised?

R4. Thank you very much for this constructive comment. We appreciate this opportunity to clarify why total iNOS protein concentration is less representative for metabolic-related inflammatory profiling. The limited correlation between iNOS protein abundance and its bioactivity primarily arises from the intrinsic structural and functional properties of iNOS itself. It is well established that iNOS enzymatic activity strictly depends on homodimer formation and only the dimeric form of iNOS is capable of catalyzing the production of NO from L-arginine. In contrast, monomeric iNOS is enzymatically inactive but remains fully detectable by conventional anti-iNOS antibodies and therefore contributes to the total protein concentrations in ELISA

assays. Moreover, cellular iNOS proteins exist in a dynamic equilibrium between inactive monomers and active dimers. The heme group, the H4B cofactor, and the substrate have all been shown to contribute to dimer stability, leading to functional inactivation despite the presence of detectable protein. In the context of EVs, this discrepancy is expected to be further pronounced. During EV biogenesis, iNOS may be packaged in different conformational or activation states. Consequently, some EVs may carry properly folded and active iNOS dimers, whereas others may carry inactive monomers or partially degraded forms. This heterogeneity leads to a decoupling between total protein abundance and the overall enzymatic output at the EV population level. In addition, the higher analytical sensitivity of the metabolic ACTIVITY assay compared with ELISA-based protein quantification (Figure 3F) allows for more precise detections, particularly at low EV concentrations. As a result, enzymatic activity can be detected even when total protein levels are close to or low the reliable range of ELISA, further enhancing its ability to reflect biologically relevant inflammatory states. However, the principal advantages of the ACTIVITY strategy lies in its selective reporting of functionally competent iNOS, rather than total protein abundance. Because iNOS activity directly mediates nitric oxide-driven metabolic and inflammatory responses in macrophages, we therefore focused on directly measuring iNOS enzymatic activity as a more accurate and biologically relevant indicator of inflammatory status, rather than relying solely on total protein concentration. We have revised the manuscript to include this mechanistic explanation and thank the reviewer again for this insightful question. The revised Main Text is shown as follow:

“This discrepancy between total iNOS protein level and its biological activity is probably due to the heterogeneity of iNOS molecules, as only a fraction exists in enzymatically active dimers, while monomers or partially folded proteins contribute to the total protein detected by ELISA [31]. In contrast, this proposed ACTIVITY method mainly relies on the enzymatic activity of EV unit, which not only provides a stronger correlation with the inflammation response in the organ, but also offers greater sensitivity enabling precise detections even at low EV concentrations. Across all clinical samples tested, the ACTIVITY assay for detecting EVs-iNOS could better reveal the inflammation in the lung.”

Reference:

[31] Kolodziejcki, P. J., Rashid, M. B. & Eissa, N. T. Intracellular formation of ‘undisruptable’ dimers of inducible nitric oxide synthase. *Proc. Natl. Acad. Sci. U. S. A.* **100**, 14263–14268 (2003).

Q5. To prove the specificity of this assay, it is needed to modulate EV iNOS expression levels by overexpression or knockout of iNOS in parent cells and then perform EV isolation and detection.

R5. We appreciate this insightful comment. We have modulated iNOS expression in the parental RAW264.7 cells by transfect pSpCas9(BB)-2A-EGFP (PX458) plasmids carrying mNos2-sgRNA-1 into RAW cells to knockout iNOS. The plasmids contain Cas9 and sgRNA sequences targeting the Nos2 gene, allowing efficient CRISPR/Cas9-mediated gene editing. Following transfection, GFP expression was observed clearly, which confirms the successful iNOS knockout (Fig. S13). The transfected cells were subsequently incubated with IL-4 to induce M2 polarization or with LPS to induce M1 polarization. EVs were isolated from these two batches of iNOS-deficient cells, and the metabolic EVs level were obtained through ACTIVITY method. As shown in the revised Figure 3N, the metabolic EVs show negligible difference and low metabolic levels. Thus, the specificity of this metabolic EVs to iNOS enzymes was

proved. The revised Main Text is shown as follows:

“To verify the essential role of metabolic EVs-iNOS in characterizing macrophage polarization, an iNOS gene knockout RAW 264.7 cell model was constructed using the CRISPR/Cas9 system (sequences provide in supplementary information). As shown in Fig. 3N, EVs derived from iNOS knockout cells treated either LPS or IL-4 exhibit consistently low metabolic EVs-iNOS levels, where the current was normalized to normal RAW264.7 derived-EVs. It further confirms the specificity of iNOS for the proposed metabolic activity of EVs.”

Figure 3. ACTIVITY for EV metabolic activity analysis. (N) Schematic illustration of iNOS knockout in RAW 264.7 cells, and corresponding ACTIVITY readout of metabolic EVs-iNOS levels in these transfected cells with different phenotypes.

Fig. S13. Fluorescence intensity of the GFP in wide-type and the transfected RAW 264.7 cells.

Table 1 Details for the plasmid

Gene name	mNos2-sgRNA-1
sg-seq	TCACAGCTCATCCGGTACGC
Resistance	Amp ⁺
Vector	pSpCas9(BB)-2A-EGFP(PX458)
5'Clone site	BbsI
3'Clone site	BbsI

Q6. The protocol of M1-like macrophage (M1) induction used in this study is not standard, and mouse macrophage cell lines (RAW 264.7 cells) were pretreated with IL-4 and then stimulated with LPS or BLZ945. In fact, LPS alone is sufficient to induce M1 macrophages, please explain the reason. Also, BLZ945 is a CSF-1R inhibitor, is there any evidence to support that it can induce M1 macrophages and what is the underlying mechanism to do so? The phenotype of macrophages used in this study should be carefully assessed, as they are unlikely classic M1 macrophages.

R6. We sincerely appreciate the reviewer's comment regarding macrophage polarization. We fully agree that the combination of LPS and IFN- γ represents a classical protocol for inducing M1 macrophages. In the present work, LPS was employed as an inflammatory stimulus to model bacterial infection-associated macrophage activation, with particular relevance to pulmonary inflammation. This approach was chosen to reflect disease relevant inflammatory conditions rather than to strictly reproduce canonical M1 differentiation under optimized in vitro settings. In addition, IL-4 pretreatment was applied to establish an initial M2-like anti-inflammatory state, thereby providing a defined baseline from which dynamic phenotypic changes could be examined following inflammatory stimulation. This design allowed us investigate macrophage plasticity and to assess whether EV-associated metabolic activity responds sensitively to shifts in inflammatory status, which is central to the aim of this study. With respect to BLZ945, we acknowledge that it is not a conventional M1-inducing agent. In this study, BLZ945 was used as a phenotype-modulating perturbation based on prior evidence that inhibition of CSF-1R signaling by BLZ945 suppresses M2 polarization and shifts macrophage phenotypes toward a more pro-inflammatory state (In Vivo. 35, 119–129 (2021)). To avoid potential ambiguity and mislead, we have revised the manuscript to clarify that LPS stimulation was used to induce an M1-like inflammatory activation, rather than to define classical M1 macrophage differentiation. We also revised the definitions of M1 and M2 macrophages in the BLZ945 treatment, as these may not accurately reflect the classical phenotypes.

The revised Main Text is shown as follows:

“To further evaluate the capability of this ACTIVITY method for quantifying macrophage polarization, we used inflammatory macrophage models stimulated with lipopolysaccharide (LPS) to induce M1-like inflammatory activation [28]. RAW 264.7 macrophages pretreated with IL-4 to establish a M2-like starting state were then incubated with LPS for varying durations. Given that CD86 and CD206 are characteristic markers of M1 and M2 macrophages, respectively, the intracellular CD86/CD206 ratio obtained from flow cytometry was used as a referential indicator for macrophage polarization (Fig. S10). As shown in Fig. 3I, the CD86/CD206 ratio increased with LPS incubation time, indicating that the macrophages polarized towards the M1 pro-inflammatory phenotype. The corresponding metabolic EVs-iNOS levels from LPS-treated macrophages were measured by isolating EVs from the culture supernatant and lysed for ACTIVITY assay. Fig. 3J showed that the relative current responses, representing metabolic EVs-iNOS levels, increased with longer LPS incubation. Similarly, BLZ945, which is known to suppresses M2 polarization, was used to regulate macrophage polarization (Fig. 3K).

We appreciate the reviewer's comment, which has helped us clarify both experimental rationale and the terminology used to describe macrophage polarization in this study.

References:

ALMAHARIQ, M. F. et al. Inhibition of Colony-Stimulating Factor-1 Receptor Enhances the Efficacy of Radiotherapy and Reduces Immune Suppression in Glioblastoma. *In Vivo*. 35, 119–129 (2021).

Q7. Cell lines may have varied biological properties compared to the primary cells, particularly in immune cells, and thus primary macrophages should be included in each test.

R7. We appreciate this constructive comment. To address this concern, we have supplemented experiments using primary macrophages differentiated from human peripheral blood monocytes, SC cells for validation. Briefly, the cells were cultured in differentiation medium and then subsequently to polarization medium to induce the SC-M1 and SC-M2 macrophages. The expression of CD86 was employed for characteristic marker for M1 polarization. EVs were isolated from these two batches of primary macrophages, and the metabolic EVs levels were obtained through the ACTIVITY method. As shown in the following data, the metabolic EV levels show distinct discrimination, with the M1 macrophage-derived EVs exhibiting higher metabolic levels than M2 EVs. Thus, the generality of this ACTIVITY method for the metabolic EVs to iNOS enzymes was further confirmed, meanwhile supporting the applications in inflammation diagnosis. The revised Main Text is shown as follow:

“To further promote this ACTIVITY method for point-of-care diagnosis of human inflammatory-related diseases, primary human monocyte-derived macrophages were used to validate the metabolic EVs-iNOS detection. A non-cancerous human monocyte model, SC cells were differentiated into primary human macrophages and subsequently polarized into M1 and M2 phenotypes by culturing with differentiation and polarization medium [29]. The expression of CD86 was employed for the characterization of M1 polarization (Fig. S15). As shown in Fig. 4O, metabolic EVs-iNOS levels in these SC-M1 macrophages clearly distinguished from those of in SC- M2 cells. Given the consistent elevated metabolic EVs-iNOS levels were consistently observed in M1 inflammatory phenotypes, including both RAW 264.7 cells and human SC-M1 macrophages, we propose that this metabolic activity of EVs can be universally used to describe macrophage polarization, offering an alternative cell-free biomarker for internal organ inflammation.”

Figure 3. ACTIVITY for EV metabolic activity analysis. (O) Schematic illustration of the differentiation and polarization of SC monocytes, and corresponding ACTIVITY readout of metabolic EVs-iNOS levels.

Q8. BALF samples contain diverse cell types, but the authors claimed that “BALF containing abundant EVs primarily derived from lung macrophages”. To support this statement, it is required to assess the exact percentages of the lung macrophages-derived EVs in the total BALF EVs.

R8. Thank you very much for pointing out this important concern. To address this issue and to provide a comparative analysis of the contribution of major cellular sources to the BALF EV pool, we performed an EV-based flow cytometry. Specifically, FITC anti-human CD4, PE anti-human CD66b, and FITC anti-human EpCAM were used for the identification and staining of macrophage-derived EVs (EVs-M), neutrophil-derived EVs (EVs-N), and epithelial cell-derived EVs (EVs-E), respectively, which represent the major EV sources in BALF. In addition, APC anti-human CD63 was employed as an intrinsic-EV marker to quantify total EVs, and the proportion of each EV subtype was calculated relative to total CD63 EVs. Through this comparative analysis, we found that macrophage-derived EVs accounted for approximately 40% of total BALF EVs population, a proportion that was notably higher than that of neutrophil-derived and epithelial cell-derived EVs. These results provide quantitative evidence supporting our statement that macrophage-derived EVs constitute a major fraction of BALF EVs, while also highlighting the relative contributions of other major cellular sources.

In addition, to evaluate the functional contribution of different EV subtypes, we conducted an immunobead-based pull-down assay prior to EV lysis and ACTIVITY measurement. Antibodies against CD4, CD66b and EpCAM were utilized to selectively isolate macrophage-derived EVs (EVs-M), neutrophil-derived EVs (EVs-N) and epithelial cell-derived EVs (EVs-E), respectively. As demonstrated in Fig. S21, macrophage-derived EVs shows the highest EV-iNOS activity among the three EV populations examined. This observation is consistent with the well-recognized functional plasticity of macrophages and their prominent role in inflammatory signaling and EV-mediated intercellular communication during lung inflammation. We noted that these results represent an initial functional comparison of major BALF EV subtypes. Further studies incorporating additional cell-specific markers and higher-resolution EV profiling approaches will be valuable for more comprehensively defining the cellular origins and functional heterogeneity of BALF EVs.

We have revised the manuscript to clarify these points and the revised Main Text is as follows:

“It is worth noting that iNOS expression may be elevated in neutrophils or other immune cells during inflammation, and BALF samples contain part of these cell-derived EVs. However, comparative analysis of EVs from neutrophils, macrophages, and endothelial cells, which represent the major EV sources, showed that macrophage-derived EVs exhibited higher EV-iNOS activity (Fig. S21). This may be attributed to the high functional plasticity of macrophages and the enhanced cellular crosstalk mediated by EVs that occurs throughout the inflammatory process. Further studies are needed into the precise mechanisms by which macrophages deliver metabolic markers via EVs to strengthen these findings.”

Fig. S21. The proportion and metabolic EV-iNOS level of macrophage-derived EVs (EVs-M, CD4⁺), neutrophil-derived EVs (EVs-N, CD66b⁺) and epithelial cell-derived EVs (EVs-E, EpCAM⁺) among EVs isolated from BALF of a pneumonia patient. EV proportion was identified by EV-based flow cytometry and each proportion was calculated relative to total CD63⁺ EVs. Metabolic EV-iNOS level was measured following an immunobeads-based pull down assay.

Q9. In a clinical translation aspect, the sample number for patients of each group is too small, and these findings need to be verified in a more large cohort of patients.

R9. Thank you very much for this important comment. We fully agree that a larger cohort of patients is essential to validate the diagnostic value of EV-associated iNOS activity. In this revision, we have increased the number of clinical samples to a total of 50 cases, including 32 patients with pneumonia and 18 healthy controls. All these newly collected samples were processed and analyzed consistently with our original methodology, and the updated results have been incorporated into the revised Figure 4, particularly Figure 4D-F. After including these additional 21 cases, the AUCs for ACTIVITY and ELISA methods are 0.97 and 0.67, respectively. These expanded data further strengthen the robustness and reliability of our findings and provide stronger evidence for the diagnostic performance of this iNOS-based metabolic EV.

The relevant updated results and figures are provided in the Main text as follows:

“We first analyzed EVs from BALF samples collected from 32 pneumonia patients and 18 healthy controls, with patient groups defined according to the integrated clinical diagnosis (see supplementary information for clinical information for these participants). Fig. 4D shows the profiling results of these clinical samples, which reflect the intrinsic EV-associated activity rather than variation in EV numbers, as all the EVs samples were pre-quantified before analysis. The normalized metabolic EVs-iNOS levels were significantly higher in the patient samples compared to the healthy controls. Scatter plots and significance analysis were performed using the relative current signals, demonstrating an excellent discrimination between healthy individuals and pneumonia patients (Fig. 4E). For the same BALF samples, we conducted a comparative analysis using the conventional ELISA method for iNOS protein detection in EVs. Receiver operator characteristic (ROC) curve analysis was utilized to evaluate the accuracy of these two methods. As demonstrated in Fig. 4F, the ACTIVITY method for metabolic EVs-iNOS exhibited an area under the curve (AUC) of 0.97, showing better performance than ELISA with an AUC of 0.67. As a comparison, a representative indicator of serum C-reactive protein (CRP) level was supplemented in Figure 4F, showing the AUC of 0.91. It demonstrated that the iNOS protein level in BALF-EVs was less representative of the

metabolic-related inflammatory profiling compared with the ACTIVITY assay. This discrepancy between total iNOS protein level and its biological activity is probably due to the heterogeneity of iNOS molecules, as only a fraction exists in enzymatically active dimers, while monomers or partially folded proteins contribute to the total protein detected by ELISA [31]. In contrast, this proposed ACTIVITY method mainly relies on the enzymatic activity of EV unit, which not only provides a stronger correlation with the inflammation response in the organ, but also offers greater sensitivity enabling precise detections even at low EV concentrations. Across all clinical samples tested, the ACTIVITY assay for detecting EVs-iNOS could better reveal the inflammation in the lung.”

Figure 4. Clinical application of the ACTIVITY strategy for pneumonia diagnostics. (D) The ACTIVITY (top) and ELISA (bottom) readouts of clinical samples, including 32 pneumonia patients and 18 healthy individuals. The signal intensities were averaged over triplicate measurements of each sample and normalized by min-max normalization. (E) Statistical analysis of the metabolic EV-iNOS levels in pneumonia patients and healthy controls. (F) ROC curves for the metabolic EV-based ACTIVITY method (red line), the iNOS-based ELISA method (black line) and the serum parameter CRP (gray line). (G) Left: Representative CT images of pneumonia patient before and after clinical treatment. Right: Evaluation of the metabolic EV-iNOS levels in BALF samples from pneumonia patients (n=4) before and after treatment. Both the ACTIVITY and ELISA data were normalized to the EV numbers.

Q10. Many important information in the methods section was missing. For example, the lysis buffer of EVs for electrochemical measurements is unclear. It seems that RIPA lysis buffer was used, however, RIPA buffer contains high concentrations of detergents, and whether they can affect the activities of the detected proteins should be determined. And thus more details must be added in the methods sections that allow the replication of this study.

R10. Thank you very much for this comment and we apologize for not clearly describing the experimental details in the original manuscript. Actually, in our study, EVs were lysed with RIPA lysis buffer for the BCA assay, which could ensure complete protein extraction. While for the iNOS ELISA and metabolic ACTIVITY measurands, EVs were lysed with a freshly prepared NP-40 buffer supplemented with 1 mM PMSF, which preserves the protein conformation and enzymatic activity. Importantly, the lysates were immediately used for enzymatic evaluation and subsequently diluted for electrochemical measurements. Because no

prolonged storage or incubation was involved after lysis, the potential impact of detergents on enzyme activity was minimized.

We also evaluate the stability of iNOS activity during EV storage. As shown in the following data, the storage of EVs at -80 °C could minimize the loss of enzyme activity. Regarding the iNOS activity after lysis, all electrochemical and ELISA measurements were performed immediately, without too much interim storage. These results confirm that iNOS activity within EVs remains stable under our experimental storage and lysis conditions.

Figure 3. (D) Biocatalytic stability of EVs during storage at -80 °C. Relative activity on a given day was normalized to freshly acquired EVs.

The revised Method section have been carefully revised as follows:

EV collection, characterization and analysis

EVs were collected from the cell culture supernatants or bronchoalveolar lavage fluid. The ethical approval of the study was obtained from the Ethics Committee of the first Affiliated Hospital of Soochow University (2023455). The case for RAW 264.7 cell-derived EVs, vesicle collection was performed after culturing cells with a vesicle-depleted medium (containing dialyzed fetal bovine serum) for 48 h. EVs were enriched from the cell culture supernatants by ultracentrifugation. Briefly, the samples were first centrifuged at 500 g for 5 min and 10000 g for 90 min to remove cells and large debris, respectively. After flited through a 0.22 μ m Millipore filter, the supernatant was then ultracentrifuged at 100,000g for 2 h to pellet EVs. The obtained EVs samples were resuspended in PBS, aliquoted and stored at -80 °C for further usage.

Morphological characterization of EVs was performed by transmission electron microscopy (TEM) with 3% uranyl acetate staining. Nanoparticle tracking analysis (NTA) was performed to demonstrate the size distributions and concentrations. After the lysis of EVs, Western blot analysis was conducted for protein characterization according to the manufacturer's protocols. For EVs collected from both cell culture supernatants and BALF samples, TEM, NTA, and Western blot analyses were performed to confirm the EV characteristics.

In the other case, for EV collections from bronchoalveolar lavage fluid, exosomes extraction kit (Beijing Baiao Laibo Technology Co. Ltd, China)) was employed after centrifugation at 8000 g for 5 min to remove sputum and other impurities. The collected EVs samples were further lysed for ELISA and electrochemical analysis. EV protein content was firstly quantified using a BCA protein assay (Beyotime Biotech, China),

after which iNOS ELISA and the ACTIVITY analysis were performed using equal EV input. This normalization allowed us to compare the intrinsic iNOS activity per EV rather than the total EV yield. For the BCA assay, EVs were lysed with RIPA lysis buffer to ensure complete protein extraction. In contrast, for iNOS ELISA and enzymatic activity analysis, EVs were lysed with a freshly prepared NP-40 buffer supplemented with 1 mM PMSF, which preserves protein conformation and enzymatic activity. For the ACTIVITY measurement, specifically, 200 μ L of lysis buffer was added to the EV precipitate, followed by incubation with 20 μ L of 5 mM L-Arg and 20 μ L of 1 mM NADPH. Then this mixture containing NO, the product of the enzymatic reaction, was transferred to 1.8 mL PBS solution in an electrochemical cell. The i-t curves were subsequently recorded for electrochemical measurements. The electrochemical readouts for metabolic EVs-iNOS were obtained after background subtraction, which accounted for the possible interference of lysis buffer and enzyme substrate in PBS. A min-max normalization was subsequently used unless otherwise stated. Regarding the detection time, EV lysis requires ~15 min, the enzymatic reaction takes ~30 min, and the electrochemical readout could be obtained within a few minutes.

Reviewer #2:

In the current manuscript, Yu et al. demonstrate that the metabolic-related inducible nitric oxide synthase (iNOS) activity of macrophage-derived EVs serves as an effective biomarker for phenotypic profiling and further evaluating lung inflammation. The authors developed an ACTIVITY method based on extracellular vesicles (EVs) diagnosis, utilizing the functional metabolic activity of iNOS in macrophage-derived EVs rather than static molecular cargo. Overall, the study is well-designed and interesting. Somehow, the study looks preliminary and there are many important concerns that need to be addressed.

RESPONSE: We greatly appreciate the reviewer's positive recommendation and thoughtful comments. Please find our detailed responses to each point below.

Q1. The 14-day stability test (Fig. 3F) solely assesses storage stability. The study overlooks mentioning the stability of defects in WS₂-DH QDs following prolonged storage or repeated uses (like performance alterations after multiple CV tests). It is advisable to present pertinent data to bolster the practical application feasibility.

R1. We appreciate the reviewer's insightful comment regarding the stability of the WS₂-DH QDs-modified electrode during repeated use. To address this, we performed multiple CV measurements on the same modified electrode repeatedly within a 12-hour period. As shown in the following data, the NO oxidation peak exhibited negligible variation after continuous time-coursing CV scans, indicating that the WS₂-DH QDs modified electrode interface remained stable without noticeable degradation or defect alteration. In our practical detection procedures, all the electrochemical measurements generally completed within 4 hours, a time frame in which the electrochemical response remains relative consistent. These results, together with the stability after 14 days of storage mentioned in the main article, demonstrate the practical application feasibility of the modified electrode over repeated use and typical sensing durations.

The revised Main Text is shown as follow:

"This could also be explained by the CV curves in Fig. 2G, where the NO oxidation potential at WS₂-DH was more negative, and its peak current was much higher than those of WS₂-DL and WS₂ DM, indicating the highest catalytic activity toward NO oxidation. Also, these modified electrodes exhibited better electrocatalytic performance than the generally used carbon-based electrodes such as reduced graphene oxide-modified electrode (Fig. S5). The stability of WS₂-DH QDs modified electrode was evaluated by multiple CV measurements, which showed a stable NO oxidation peak with the relative standard deviation (RSD) about 3.3% over 12-hour duration at 2-hours intervals (Fig. S6).

Fig. S6. CV measurements of the WS₂-DH QDs modified electrode in PBS electrolyte with 0.18 mM NO over a 12-hour period, recorded at 2-hour intervals.

Q2. Potential Contamination Concerns in EV Preparations: The ultracentrifugation protocol (100,000g for 2h) may inadvertently co-isolate apoptotic bodies or protein aggregates, particularly in BALF samples. Data pertaining to EV-specific markers (e.g., CD63, TSG101) are absent (Fig. 3B, S5).

R2. Thank you very much for this constructive comment. In response to the concern regarding potential co-isolation of protein aggregates, we have supplemented additional characterization experiments. First, we performed Western blot analysis to confirm the presence of characteristic EV markers, including CD9, CD63, and TSG101 (revised Figure 1C), supporting the successful enrichment of EVs.

Figure 1. (C) TEM image and Western blot analysis of macrophage-derived EVs. scale bar, 100 nm.

In addition, we further purified the obtained EVs samples using iodixanol density gradient ultracentrifugation (DGUC) to assess potential contamination following the MISEV2023 guidelines. The resulting particle fractions were quantified with resistive pulse sensing method and the encapsulated iNOS protein were quantified via ELISA method after a vesicle lysis process. Specifically, obvious four fractions of particles were observed, with F4 in the bottom of tube showing as protein aggregates. By normalizing the iNOS protein concentrations to particles numbers, layer 4 shows the minimum iNOS concentration. In other words, potential proteins contaminants are expected to contribute negligibly to iNOS sensing. Nonetheless, we sincerely appreciate the reviewer's reminder, and we will adopt more rigorous EV isolation protocols in our future studies to further ensure the specificity and reliability of our findings. We also add limitation discussions in the Main Text as follows:

“Meanwhile, there are some limitations in this study. Such as, we mainly focus on the bulk EV analysis,

which may overlook the heterogeneity of EV subpopulations presented in biofluids, including both inter- and intra-subpopulation heterogeneity [32]. First, immunobeads mediated pull down-method could be employed to isolate EVs from specific cell types, which highly depends on the specificity and affinity of antibodies. Second, density-gradient centrifugation and size-exclusion chromatography can improve EV purity and enable the separation of EVs with different size distributions, while sacrificing the EV production in each subtype, thereby requiring larger sample volumes or more sensitive detection methods. With technological advances, integration of electrochemical platform with single entity approaches could enable subtype-specific EV profiling, which may offer higher sensitivity and more detailed insights into metabolic activity at the single vesicle level. In addition, macrophages exhibit functional plasticity in the lung, and their responses to the inflammatory microenvironment are dependent on the specific pulmonary disease context. Future single-vesicle analysis that access the heterogeneity of EV metabolic signatures may therefore provide fundamental insights into the association between EV function and the detailed disease progression.”

Supplementary Figure (A) Standard curve for iNOS quantification using ELISA. (B) Four main particle fractions with F4 representing as protein aggregates. The number of particles in each fraction were obtained through resistive pulse sensing method and each iNOS concentrations were determined by ELISA. (C) iNOS concentration normalized to particle number for each fraction.

The Method Section has been detailed as follows:

“The case for RAW 264.7 cell-derived EVs, vesicle collection was performed after culturing cells with a vesicle-depleted medium (containing dialyzed fetal bovine serum) for 48 h. EVs were enriched from the cell culture supernatants by ultracentrifugation. Briefly, the samples were first centrifuged at 500 g for 5 min and 10000 g for 90 min to remove cells and large debris, respectively. After filtered through a 0.22 μm Millipore filter, the supernatant was then ultracentrifuged at 100,000g for 2 h to pellet EVs. The obtained EVs samples were resuspended in PBS, aliquoted and stored at -80 °C for further usage.”

Q3. For experiments involving iNOS in this study, it is advisable to employ NOS2 gene knockout mice as controls to affirm the study's accuracy.

R3. This is a great point and thank you so much for this comment. To verify the specificity of our ACTIVITY method for iNOS, we employed NOS2 gene knockout macrophages of mice as an in vitro model for

controls. Specifically, RAW264.7 cells were transfected by pSpCas9(BB)-2A-EGFP (PX458) plasmids carrying mNos2-sgRNA-1 to knockout iNOS gene. The plasmids contain Cas9 and sgRNA sequences targeting the Nos2 gene, allowing efficient CRISPR/Cas9-mediated gene editing. Following transfection, GFP expression was observed clearly, which confirms the successful iNOS knockout (Fig. S13). The transfected cells were subsequently incubated with IL-4 to induce M2 polarization or with LPS to induce M1 polarization. EVs were isolated from these two batches of iNOS-deficient cells, and the metabolic EVs level were obtained through ACTIVITY method. As shown in the revised Figure 3N, the metabolic EVs show negligible difference and low metabolic levels. Thus, the specificity of this metabolic EVs to iNOS enzymes was proved.

The revised Main Text is shown as follows:

“To verify the essential role of metabolic EVs-iNOS in characterizing macrophage polarization, an iNOS gene knockout RAW 264.7 cell model was constructed using the CRISPR/Cas9 system (sequences provide in supplementary information). As shown in Fig. 3N, EVs derived from iNOS knockout cells treated either LPS or IL-4 exhibit consistently low metabolic EVs-iNOS levels, where the current was normalized to normal RAW264.7 derived-EVs. It further confirms the specificity of iNOS for the proposed metabolic activity of EVs.”

Figure 3. ACTIVITY for EV metabolic activity analysis. (N) Schematic illustration of iNOS knockout in RAW 264.7 cells, and corresponding ACTIVITY readout of metabolic EVs-iNOS levels in these transfected cells with different phenotypes.

Fig. S13. Fluorescence intensity of the GFP in wide-type and the transfected RAW 264.7 cells.

Table 1 Details for the plasmid

Gene name	mNos2-sgRNA-1
sg-seq	TCACAGCTCATCCGGTACGC
Resistance	Amp ⁺
Vector	pSpCas9(BB)-2A-EGFP(PX458)
5'Clone site	BbsI

Q4. The study presumes iNOS activity remains constant throughout EV storage (-80°C) and lysis. No data are provided regarding time- or temperature-dependent activity loss.

R4. We appreciate the reviewer's valuable comment regarding the stability of iNOS activity during EV storage. To address this concern, we assessed the biocatalytic activity of iNOS within EVs under storage at - 80 °C for different durations, and subsequently lysed under the same conditions used in our assays. The iNOS activity was determined, which shows nearly unchanged throughout such storage period.

The revised Main Text is shown as follow:

“These results demonstrate that the disruption of EVs structure is essential for iNOS sensing, strongly suggesting that iNOS is localized within the lumen of EVs rather than on the surface. For this reason, iNOS exhibited considerable stability during EV storage, with its biocatalytic activity remained nearly unchanged over 42 days at -80 °C (Fig. 3D). This enhanced stability is probability attributed to the inside localization of iNOS, where lipid encapsulation provides effective protection during the long-term storage.”

Figure 3. (D) Biocatalytic stability of EVs during storage at -80 °C. Relative activity on a given day was normalized to freshly acquired EVs.

Q5. LPS was utilized to stimulate M1-type macrophage polarization in this study, whereas the conventional method for inducing M1-type macrophage polarization involves LPS combined with IFN-γ. The authors should further refine their research protocol.

R5. We sincerely thank the reviewer's comment regarding the macrophage polarization protocol. We fully agree that the LPS + IFN-γ combination represents the classic protocol to induce M1 macrophages both

in in vitro and in vivo. In the present work, LPS was employed as an inflammatory stimulus to model bacterial infection-associated macrophage activation, with the aim of capturing a pro-inflammatory state relevant to infection-associated lung inflammation, rather than to strictly reproduce canonical M1 differentiation under optimized cytokine conditions. In addition to the classical LPS and IFN- γ protocol, LPS stimulation alone has also been employed in previous studies to induce an M1-like pro-inflammatory activation in macrophages, with increased iNOS expression. In this context, LPS was therefore employed as a simplified and biologically relevant inflammatory trigger to examine macrophage responses under infection-related pulmonary inflammatory conditions. Accordingly, our experimental design focused on establishing an inflammatory macrophage activation state representative of the lung inflammatory microenvironment and on assessing whether EV-associated metabolic activity responds sensitively to such activation. To avoid potential ambiguity and mislead, we have revised the manuscript to clarify that LPS stimulation was used to induce an M1-like inflammatory activation, rather than to define classical M1 macrophage differentiation:

“To further evaluate the capability of this ACTIVITY method for quantifying macrophage polarization, we used inflammatory macrophage models stimulated with lipopolysaccharide (LPS) to induce M1-like inflammatory activation [28].”

We appreciate the reviewer’s comment, which has helped us refine the description and positioning of the macrophage polarization strategy in this study.

References:

Lu, G., Zhang, R., Geng, S. et al. Myeloid cell-derived inducible nitric oxide synthase suppresses M1 macrophage polarization. *Nat Commun* 6, 6676 (2015).

Y. Zhang, X. Li, Z. Luo, L. et al. ECM1 is an essential factor for the determination of M1 macrophage polarization in IBD in response to LPS stimulation. *Proc. Natl. Acad. Sci.* 2020, 117, 3083–3092.

Q6. Currently, only the RAW 264.7 cell line is used. Incorporating experimental results from primary macrophages (like human peripheral blood mononuclear cell-derived macrophages or BMDM) would bolster the conclusions' universality.

R6. Thank you very much for this constructive comment. To address this concern, we have supplemented experiments using primary macrophages differentiated from human peripheral blood monocytes, SC cells for validation. Briefly, the cells were cultured in differentiation medium and then subsequently to polarization medium to induce the SC-M1 and SC-M2 macrophages. The expression of CD86 was employed for characteristic marker for M1 polarization. EVs were isolated from these two batches of primary macrophages, and the metabolic EVs levels were obtained through the ACTIVITY method. As shown in the following data, the metabolic EV-iNOS show distinct discrimination, with the M1 macrophage-derived EVs exhibiting higher metabolic levels than M2 EVs. Thus, the generality of this ACTIVITY method for the metabolic EVs to iNOS enzymes was further confirmed, meanwhile supporting the applications in inflammation diagnosis.

The revised Main Text is shown as follow:

“To further promote this ACTIVITY method for point-of-care diagnosis of human inflammatory-related diseases, primary human monocyte-derived macrophages were used to validate the metabolic EVs-iNOS detection. A non-cancerous human monocyte model, SC cells were differentiated into primary human macrophages and subsequently polarized into M1 and M2 phenotypes by culturing with differentiation and polarization medium [29]. The expression of CD86 was employed for the characterization of M1 polarization (Fig. S15). As shown in Fig. 4O, metabolic EVs-iNOS levels in these SC-M1 macrophages clearly distinguished from those of in SC- M2 cells. Given the consistent elevated metabolic EVs-iNOS levels were consistently observed in M1 inflammatory phenotypes, including both RAW 264.7 cells and human SC-M1 macrophages, we propose that this metabolic activity of EVs can be universally used to describe macrophage polarization, offering an alternative cell-free biomarker for internal organ inflammation.”

Figure 3. ACTIVITY for EV metabolic activity analysis. (O) Schematic illustration of the differentiation and polarization of SC monocytes, and corresponding ACTIVITY readout of metabolic EVs-iNOS levels.

Q7. The patient group (n=18) and control group (n=11) sample sizes are relatively modest, and pneumonia subtypes (bacterial/viral/non-infectious) or severity stratification are unspecified. Expanding the sample size and performing stratified analysis would validate the method's broad applicability.

R7. Thank you very much for this important comment. We fully agree that a larger cohort of patients is essential to validate the diagnostic value of EV-associated iNOS activity. In this revision, we have increased the number of clinical samples to a total of 50 cases, including 32 patients with pneumonia and 18 healthy controls. All these newly collected samples were processed and analyzed using the same experimental protocols and analytical criteria as those applied in the original cohort, ensuring full methodological consistency. The updated results have now been incorporated into the revised Figure 4, particularly Figure 4D-F. After including these additional samples, the ACTIVITY assay maintained excellent diagnostic performance, yielding an AUC of 0.97, whereas the EV-based iNOS ELISA showed an AUC of 0.67. These results further strengthen the robustness and reliability of our findings and provide stronger evidence for the diagnostic performance of this iNOS-based metabolic EV.

The relevant updated results and figures are provided in the Main text as follows:

“We first analyzed EVs from BALF samples collected from 32 pneumonia patients and 18 healthy controls, with patient groups defined according to the integrated clinical diagnosis (see supplementary information for clinical information for these participants). Fig. 4D shows the profiling results of these clinical samples, which reflect the intrinsic EV-associated activity rather than variation in EV numbers, as all the EVs samples were pre-quantified before analysis. The normalized metabolic EVs-iNOS levels were significantly higher in the patient samples compared to the healthy controls. Scatter plots and significance analysis were performed using the relative current signals, demonstrating an excellent discrimination between healthy individuals and pneumonia patients (Fig. 4E). For the same BALF samples, we conducted a comparative analysis using the conventional ELISA method for iNOS protein detection in EVs. Receiver operator characteristic (ROC) curve analysis was utilized to evaluate the accuracy of these two methods. As demonstrated in Fig. 4F, the ACTIVITY method for metabolic EVs-iNOS exhibited an area under the curve (AUC) of 0.97, showing better performance than ELISA with an AUC of 0.67. As a comparison, a representative indicator of serum C-reactive protein (CRP) level was supplemented in Figure 4F, showing the AUC of 0.91. It demonstrated that the iNOS protein level in BALF-EVs was less representative of the metabolic-related inflammatory profiling compared with the ACTIVITY assay. This discrepancy between total iNOS protein level and its biological activity is probably due to the heterogeneity of iNOS molecules, as only a fraction exists in enzymatically active dimers, while monomers or partially folded proteins contribute to the total protein detected by ELISA [31]. In contrast, this proposed ACTIVITY method mainly relies on the enzymatic activity of EV unit, which not only provides a stronger correlation with the inflammation response in the organ, but also offers greater sensitivity enabling precise detections even at low EV concentrations. Across all clinical samples tested, the ACTIVITY assay for detecting EVs-iNOS could better reveal the inflammation in the lung.”

Figure 4. Clinical application of the ACTIVITY strategy for pneumonia diagnostics. (D) The ACTIVITY (top) and ELISA (bottom) readouts of clinical samples, including 32 pneumonia patients and 18 healthy individuals. The signal intensities were averaged over triplicate measurements of each sample and normalized by min-max normalization. (E) Statistical analysis of the metabolic EV-iNOS levels in pneumonia patients and healthy controls. (F) ROC curves for the metabolic EV-based ACTIVITY method (red line), the iNOS-based ELISA method (black line) and the serum parameter CRP (gray line). (G) Left: Representative CT images of pneumonia patient before and after clinical treatment. Right: Evaluation of the metabolic EV-iNOS levels in BALF samples

from pneumonia patients (n=4) before and after treatment. Both the ACTIVITY and ELISA data were normalized to the EV numbers.

We acknowledge that the current cohort size remains insufficient to support statistically powered subgroup analyses based on pneumonia etiology (e.g., bacterial vs viral) or disease severity. At this stage, our primary objective was to establish proof-of-concept that EV-associated iNOS enzymatic activity in BALF can serve as a functional biomarker reflecting lung inflammatory status, rather than to delineate disease etiology or severity-dependent differences. Accordingly, we focused on comparing clinically diagnosed pneumonia patients with healthy controls to validate the overall diagnostic performance of the ACTIVITY assay. We have revised the manuscript to explicitly acknowledge this limitation and to clarify that future studies with larger, well-characterized cohorts and detailed clinical stratification will be necessary to determine whether EV-associated iNOS activity can further discriminate pneumonia subtypes or correlate with disease severity. The following discussion has been added (Page 7):

“While the present study focused on establishing EV-associated iNOS activity as a general marker of lung inflammation, larger cohorts with detailed etiological and severity stratification will be required to assess its specificity across different pneumonia subtypes.”

Q8. Imaging diagnosis serves as the gold standard in the article, yet it lacks comparison with existing inflammatory markers (such as serum CRP, PCT). Correlation analysis should be incorporated to elucidate its additional diagnostic merit.

R8. Thank you very much for this valuable comment. Following the reviewer's suggestion, we have added a comparison with an established inflammatory marker. In this study, pneumonia was determined based on chest imaging (X-ray or CT) together with clinical symptoms, in accordance with routine clinical practice. This integrated clinical confirm therefore served as the diagnostic reference standard. As suggested, we included C-reactive protein (CRP) as a representative inflammatory marker for comparison. ROC analysis for CRP has been added to Figure 4F, yield an AUC of 0.91 in our cohort. Under the same clinical reference, the ACTIVITY assay achieved an AUC of 0.97, indicating improved diagnostic discrimination compared with both EV-based iNOS ELISA and CRP. The observed difference between the two readouts is consistent with their biological characteristics. CRP reflects systemic inflammation and is widely used as a supportive laboratory indicator, but it is neither specific to lung pathology nor a stand-alone diagnostic marker for pneumonia. In contrast, the ACTIVITY assay measures local inflammatory activity in the lung through BALF-derived EV-associated metabolic activity, providing more direct information on pulmonary inflammation.

The main text has been revised as follows:

“We first analyzed EVs from BALF samples collected from 32 pneumonia patients and 18 healthy controls, with patient groups defined according to the integrated clinical diagnosis (see supplementary information for clinical information for these participants). Fig. 4D shows the profiling results of these clinical samples, which reflect the intrinsic EV-associated activity rather than variation in EV numbers, as all the EVs

samples were pre-quantified before analysis. The normalized metabolic EVs-iNOS levels were significantly higher in the patient samples compared to the healthy controls. Scatter plots and significance analysis were performed using the relative current signals, demonstrating an excellent discrimination between healthy individuals and pneumonia patients (Fig. 4E). For the same BALF samples, we conducted a comparative analysis using the conventional ELISA method for iNOS protein detection in EVs. Receiver operator characteristic (ROC) curve analysis was utilized to evaluate the accuracy of these two methods. As demonstrated in Fig. 4F, the ACTIVITY method for metabolic EVs-iNOS exhibited an area under the curve (AUC) of 0.97, showing better performance than ELISA with an AUC of 0.67. As a comparison, a representative indicator of serum C-reactive protein (CRP) level was supplemented in Figure 4F, showing the AUC of 0.91.”

Figure 4. Clinical application of the ACTIVITY strategy for pneumonia diagnostics. (F) ROC curves for the metabolic EV-based ACTIVITY method (red line), the iNOS-based ELISA method (black line) and the serum parameter CRP (gray line).

Q9. Numerous studies indicate that iNOS can be produced by various immune cells, including neutrophils, dendritic cells, and MDSCs. How this approach mitigates this clinical application limitation remains to be addressed.

R9. Thank you so much for this constructive comment. To address this issue, we supplemented the proportion of macrophage-derived EVs (EVs-M, CD4+), neutrophil-derived EVs (EVs-N, CD66b+) and epithelial cell-derived EVs (EVs-E, EpCAM+) to total BALF EVs. As shown in the revised Figure S20, macrophage-derived EVs accounted for approximately 40% of total BALF EVs. This proportion was notably higher than that of neutrophil-derived and epithelial cell-derived EVs (Fig. S21). In addition to differences in abundance, we assessed the functional contribution of these EV subtypes by measuring their EV-associated iNOS activity. Among the three populations examined, macrophage-derived EVs consistently exhibited the highest EV-iNOS activity, whereas neutrophil- and epithelial-derived EVs showed substantially lower activity levels. These results indicate that, although iNOS expression is not exclusive to macrophages, macrophage-derived EVs represent the dominant functional source of EV-associated iNOS activity in BALF under the conditions studied. This functional predominance is consistent with the well-recognized plasticity of macrophages and their central role in inflammatory signaling and EV-mediated intercellular communication during lung inflammation. Importantly, our approach does not assume macrophage exclusivity but instead relies on relative abundance and functional dominance to mitigate the

limitation posed by iNOS expression in other immune cell types. We note that this analysis represents an initial exploratory assessment of EV cellular origin and function, and further studies are needed to strengthen this claim. We have revised the Main Text and claim this limitation as follows:

“It is worth noting that iNOS expression may be elevated in neutrophils or other immune cells during inflammation, and BALF samples contain part of these cell-derived EVs. However, comparative analysis of EVs from neutrophils, macrophages, and endothelial cells, which represent the major EV sources, showed that macrophage-derived EVs exhibited higher EV-iNOS activity (Fig. S21). This may be attributed to the high functional plasticity of macrophages and the enhanced cellular crosstalk mediated by EVs that occurs throughout the inflammatory process. Further studies are needed into the precise mechanisms by which macrophages deliver metabolic markers via EVs to strengthen these findings. Meanwhile, there are some limitations in this study. Such as, we mainly focus on the bulk EV analysis, which may overlook the heterogeneity of EV subpopulations presented in biofluids, including both inter- and intra-subpopulation heterogeneity [32]. First, immunobeads mediated pull down-method could be employed to isolate EVs from specific cell types, which highly depends on the specificity and affinity of antibodies. Second, density-gradient centrifugation and size-exclusion chromatography can improve EV purity and enable the separation of EVs with different size distributions, while sacrificing the EV production in each subtype, thereby requiring larger sample volumes or more sensitive detection methods. With technological advances, integration of electrochemical platform with single entity approaches could enable subtype-specific EV profiling, which may offer higher sensitivity and more detailed insights into metabolic activity at the single vesicle level. In addition, macrophages exhibit functional plasticity in the lung, and their responses to the inflammatory microenvironment are dependent on the specific pulmonary disease context. Future single-vesicle analysis that access the heterogeneity of EV metabolic signatures may therefore provide fundamental insights into the association between EV function and the detailed disease progression.”

Fig. S21. The proportion and metabolic EV-iNOS level of macrophage-derived EVs (EVs-M, CD4⁺), neutrophil-derived EVs (EVs-N, CD66b⁺) and epithelial cell-derived EVs (EVs-E, EpCAM⁺) among EVs isolated from BALF of a pneumonia patient. EV proportion was identified by EV-based flow cytometry and each proportion was calculated relative to total CD63⁺ EVs. Metabolic EV-iNOS level was measured following an immunobeads-based pull down assay.

Reviewer #3:

Novelty- less studies have been performed on metabolic profiles of EVs compared to proteomic or genomic analysis of EVs. This is correct. The arginine based “activity probe” appears to detect iNOS metabolic activity with sensitivity when incubated with the EVs. Personally i dont know of a paper where an "ACTIVITY" probe was used to determine NOS activity in EVs. From that perspective the work has potential significance.

RESPONSE: We greatly appreciate the reviewer’s positive recommendation and thoughtful comments. Please find the following specific points.

Q1. General concerns: Some of the writing and comments throughout need rewriting

Inflammation is referred to very generally throughout, given the complexity of the macrophage EV driven inflammation which differs across each distinct lung disease i'd be cautious of general marks also in terms of polarisation. M1 like macrophage cytokines have been associated with pneumonia but the idea of pro-inflammatory M1 macrophages and M2 macrophages suppressing inflammation is likely a simplification as macrophages in lung exhibit plasticity in response to different stimuli.

Specific examples

... from L-arginine to generate line 60 missing word end of sentence

'In the inflammation in internal organs such as lung'.... Line 60 badly phrased

Approach for evaluating internal organ inflammation- general remark -line 68.

R1. Thank you very much for your valuable comments on clarity and precision. We have carefully revised the relevant parts of the manuscript to improve readability and eliminate overgeneralized claims. The revised Main text is shown as follow:

“... Specifically, M1 macrophages are characterized by elevated levels of inducible nitric oxide synthase (iNOS), a metabolic enzyme synthesizing nitric oxide (NO) from L-arginine [16]. In the inflammatory conditions affecting internal organs such as lung, direct isolation of the macrophages for analysis is challenging, and conventional diagnostic methods generally involve imaging techniques such as magnetic resonance imaging and chest computed tomography.....Therefore, directly assessing the iNOS activity of macrophage-derived EVs, which reflects the dynamic metabolic state of the parental macrophages, represents a promising diagnostic approach for evaluating inflammation in internal organs.”

Regarding macrophage polarization, we have revised the writings and highlighted the complexity and plasticity of macrophage phenotypes in the lung environment, rather than broadly referring to inflammation.

Q2. Figure 1: schematic looks ok except NOS blot not very convincing, many bands on it.

R2. Thank you very much for pointing this out. The western blot data have been replaced by a clear version and other EV-related proteins including CD9, CD 63, and Tsg101 were employed for characterization. This improvement provides a more convincing representation of iNOS expression in the EV samples. The revised Figures are shown as follow:

Figure 1. ACTIVITY-based analysis of EV metabolic dynamics. (C) TEM image and Western blot analysis of macrophage-derived EVs. scale bar, 100 nm.

Q3. Figure 2: Design and evaluation of ACTIVITY probes- no comment as not my area.

R3. No further revision was required.

Q4. Figure 3: Why are the authors using RAW 264.7 cells from mice to see if EV metabolic signature characterizes macrophage polarization? That would make sense if they were going to validate approach in a mouse after but human BALF samples are used for further validation. Is there not a more relevant model, alveolar macrophage, even macrophages from derived human PBMCs (although blood and airway macrophages differ).

R4. Thank you very much for pointing this out. To address this concern, we have supplemented experiments using primary macrophages differentiated from human peripheral blood monocytes, SC cells for validation. Briefly, the cells were cultured in differentiation medium and then subsequently to polarization medium to induce the SC-M1 and SC-M2 macrophages. The expression of CD86 was employed for characteristic marker for M1 polarization. EVs were isolated from these two batches of primary macrophages, and the metabolic EVs levels were obtained through the ACTIVITY method. As shown in the following data, the metabolic EV-iNOS show distinct discrimination, with the M1 macrophage-derived EVs exhibiting higher metabolic levels than M2 EVs. Thus, the generality of this ACTIVITY method for the metabolic EVs to iNOS enzymes was further confirmed, meanwhile supporting the applications in inflammation diagnosis.

The revised Main Text is shown as follow:

“To further promote this ACTIVITY method for point-of-care diagnosis of human inflammatory-related diseases, primary human monocyte-derived macrophages were used to validate the metabolic EVs-iNOS detection. A non-cancerous human monocyte model, SC cells were differentiated into primary human macrophages and subsequently polarized into M1 and M2 phenotypes by culturing with differentiation and polarization medium [29]. The expression of CD86 was employed for the characterization of M1 polarization (Fig. S15). As shown in Fig. 4O, metabolic EVs-iNOS levels in these SC-M1 macrophages

clearly distinguished from those of in SC- M2 cells. Given the consistent elevated metabolic EVs-iNOS levels were consistently observed in M1 inflammatory phenotypes, including both RAW 264.7 cells and human SC-M1 macrophages, we propose that this metabolic activity of EVs can be universally used to describe macrophage polarization, offering an alternative cell-free biomarker for internal organ inflammation.”

Figure 3. ACTIVITY for EV metabolic activity analysis. (O) Schematic illustration of the differentiation and polarization of SC monocytes, and corresponding ACTIVITY readout of metabolic EVs-iNOS levels.

Q5. *Figure 4: Clinical details: Are there details re the 18 pneumonia BALF samples and patient characteristics? What state of disease progression etc?*

R5. Thank you very much for this valuable comment. In the revised manuscript, we have added a summary table of available patient characteristics in the Supplementary Information, including case number, age distribution, sex, and clinical diagnosis for the full cohort (32 pneumonia patients and 18 non-pneumonia controls). Pneumonia diagnosis was determined based on integrated clinical assessment, including chest X-ray or CT imaging together with clinical symptoms, in accordance with routine clinical practice. Detailed stratification by disease stage, severity, or progression was not performed in the present study. This is primarily due to the limited cohort size and the heterogeneity of clinical presentations, which would preclude statistically meaningful subgroup analyses and risk overinterpretation. Accordingly, the current study was designed to establish proof-of-concept that EV-associated iNOS activity in BALF reflects lung inflammatory status, rather than to evaluate stage- or progression-dependent differences. In addition, following the reviewer’s suggestion, we have supplemented serum C-reactive protein (CRP) data as a representative systemic inflammatory marker. Summary statistics for CRP are included in the Table 2, and ROC analysis for CRP has been added to Figure 4F, yielding an AUC of 0.91. CRP was included as a comparative reference marker rather than as a diagnostic criterion, as it reflects systemic inflammation and is not specific to lung pathology.

The revised Main Text and supplemented clinical information are shown as follows:

“We first analyzed EVs from BALF samples collected from 32 pneumonia patients and 18 healthy controls, with patient groups defined according to the integrated clinical diagnosis (see supplementary information for clinical information for these participants) As demonstrated in Fig. 4F, the ACTIVITY method for

metabolic EVs-iNOS exhibited an area under the curve (AUC) of 0.97, showing better performance than ELISA with an AUC of 0.67. As a comparison, a representative indicator of serum C-reactive protein (CRP) level was supplemented in Figure 4F, showing the AUC of 0.91.”

Figure 4. Clinical application of the ACTIVITY strategy for pneumonia diagnostics. (F) ROC curves for the metabolic EV-based ACTIVITY method (red line), the iNOS-based ELISA method (black line) and the serum parameter CRP (gray line).

Table 2 Clinical information of cohorts involved in this ACTIVITY-based pneumonia diagnosis

Characteristic	Pneumonia	Non-pneumonia	Total
Case	32	18	50
Age			
Median	51	37	43
Range	21 - 83	13 - 56	13 - 83
Sex			
Male	21 (66%)	7 (39%)	28 (56%)
Female	11 (44%)	11 (61%)	22 (44%)
Serum CRP marker (mg/L)			
Median	73.8	3.55	28.62
Range	1.72 - 280.32	0.5 - 40.15	1.72- 280.32

Q6. Writing: Re the sentence 'BALF is widely recognized as an ideal less-invasive fluid for rapid disease evaluation....' BALF is not widely regarded as less invasive as it is obtained via bronchoscopy which is invasive, i'd rephrase. BALF samples are often obtained via a biobanking program, is this the case here?

R6. We agree with the reviewer and have revised the sentence to avoid the misleading term "less-invasive." The new version shown as:

"BALF provides direct access to the lung microenvironment for evaluation of pulmonary disease."

We also clarified that the BALF samples in this study were obtained from a hospital biobank under ethical approval and patient consent.

Q7. Figure 4 part B-TEM of BALF EVs could be better- 4B, is it from healthy or pneumonia donors?

Normally both shown.

Part C- assume this is by NTA, id state it Is it a representative of a graph from 1 healthy or 1 pneumonia patient?

Further comments, the authors have shown there is more metabolic NOS EV activity in BALF from pneumonia patients which might be promising with further characterisation.

R7. Thank you very much for pointing this out and sorry we did not clarify this in the original submission. Both the TEM and NTA characterizations in the original Figure 4 were obtained from a healthy donor. Also, we supplemented TEM and NTA characterizations of BALF-EV from a pneumonia patient to allow direct comparison between groups. As shown in the following Figures, EVs from healthy and pneumonia donors both show similar morphology and size distributions. The revised Main Text is shown as follows:

“BALF is widely recognized to provide a direct window for rapid lung disease evaluation, containing abundant EVs primarily derived from lung macrophages [30]. TEM characterization (Fig. 4B) and size distribution analysis (Fig. 4C) confirmed the abundant presence of EVs in BALF samples. Western blot analysis further verified the presence of iNOS in BALF-EVs collected from a pneumonia patient (Fig. S16). Although the morphology of EVs isolated from patient was comparable to that of healthy donor (Fig. S18), a slightly higher concentration was observed in the patient sample.”

Figure 4. Clinical application of the ACTIVITY strategy for pneumonia diagnostics. Representative TEM image (B) and particle size distribution (C) of EVs isolated from BALF of a pneumonia patient.

Fig. S18 Representative TEM image (A) and particle size distribution (B) of EVs isolated from BALF of a health donor.

Q8. In addition to clinical info what other characteristics do these EVs have, are there generally more EVs by NTA in the pneumonia patients for example?

R8. Thank you for this insightful comment. In addition to the clinical information, we also characterized

EVs derived from both pneumonia patient and healthy individual using TEM and NTA. As shown in Figure S17, NTA results revealed that BALF samples from pneumonia patients generally exhibited a modest increase in total EV concentrations compared with healthy controls (Fig. S18), which likely reflects enhanced inflammatory activity in the lung microenvironment. Importantly, to ensure that differences in iNOS activity were not confounded by variations in EV abundance, all the ACTIVITY and ELISA measurements were normalized to EV quantity. Specifically, EV samples were first quantified using a BCA protein assay, followed by iNOS ELISA measurement and the ACTIVITY analysis using equal EV input. This normalization allowed us to compare the intrinsic iNOS activity per EV rather than the total EV yield.

The revised Main Text is shown as follow:

“Although the morphology of EVs isolated from patient was comparable to that of healthy donor (Fig. S18), a slightly higher concentration was observed in the patient sample. Thus, we deduce that the metabolic profiling of EVs-iNOS in this fluid would possibly be used for the diagnosis of lung inflammation. BALF samples were first centrifuged to remove impurities, and the resulting EVs were subsequently collected, lysed, and subjected to ACTIVITY assaying to determine the metabolic EVs-iNOS levels. We first analyzed EVs from BALF samples collected from 32 pneumonia patients and 18 healthy controls, with patient groups defined according to the integrated clinical diagnosis (see supplementary information for clinical information for these participants). Fig. 4D shows the profiling results of these clinical samples, which reflect the intrinsic EV-associated activity rather than variation in EV numbers, as all the EVs samples were pre-quantified before analysis.”

Q9. How does this functionally relate back to macrophages, BALF contains EVs of many cell types including EVs from epithelial cells, neutrophils etc which may also can have elevated NOS activity. If macrophages were isolated from BALF and their ACTIVITY measured this would be more convincing.

R9. Thank you so much for this constructive comment. To address this issue, we supplemented the proportion of macrophage-derived EVs (EVs-M, CD4+), neutrophil-derived EVs (EVs-N, CD66b+) and epithelial cell-derived EVs (EVs-E, EpCAM+) to total BALF EVs. As shown in the revised Figure S20, macrophage-derived EVs accounted for approximately 40% of total BALF EVs. This proportion was notably higher than that of neutrophil-derived and epithelial cell-derived EVs (Fig. S21). In addition to differences in abundance, we assessed the functional contribution of these EV subtypes by measuring their EV-associated iNOS activity. Among the three populations examined, macrophage-derived EVs consistently exhibited the highest EV-iNOS activity, whereas neutrophil- and epithelial-derived EVs showed substantially lower activity levels. These results indicate that, although iNOS expression is not exclusive to macrophages, macrophage-derived EVs represent the dominant functional source of EV-associated iNOS activity in BALF under the conditions studied. This functional predominance is consistent with the well-recognized plasticity of macrophages and their central role in inflammatory signaling and EV-mediated intercellular communication during lung inflammation. Importantly, our approach does not assume macrophage exclusivity but instead relies on relative abundance and functional dominance to mitigate the limitation posed by iNOS expression in other immune cell types. We note that this analysis represents an initial exploratory assessment of EV cellular origin and function, and further studies are needed to

strengthen this claim. We have expanded the discussion in the main text as follows:

“It is worth noting that iNOS expression may be elevated in neutrophils or other immune cells during inflammation, and BALF samples contain part of these cell-derived EVs. However, comparative analysis of EVs from neutrophils, macrophages, and endothelial cells, which represent the major EV sources, showed that macrophage-derived EVs exhibited higher EV-iNOS activity (Fig. S21). This may be attributed to the high functional plasticity of macrophages and the enhanced cellular crosstalk mediated by EVs that occurs throughout the inflammatory process. Further studies are needed into the precise mechanisms by which macrophages deliver metabolic markers via EVs to strengthen these findings. Meanwhile, there are some limitations in this study. Such as, we mainly focus on the bulk EV analysis, which may overlook the heterogeneity of EV subpopulations presented in biofluids, including both inter- and intra-subpopulation heterogeneity [32]. First, immunobeads mediated pull down-method could be employed to isolate EVs from specific cell types, which highly depends on the specificity and affinity of antibodies. Second, density-gradient centrifugation and size-exclusion chromatography can improve EV purity and enable the separation of EVs with different size distributions, while sacrificing the EV production in each subtype, thereby requiring larger sample volumes or more sensitive detection methods. With technological advances, integration of electrochemical platform with single entity approaches could enable subtype-specific EV profiling, which may offer higher sensitivity and more detailed insights into metabolic activity at the single vesicle level. In addition, macrophages exhibit functional plasticity in the lung, and their responses to the inflammatory microenvironment are dependent on the specific pulmonary disease context. Future single-vesicle analysis that access the heterogeneity of EV metabolic signatures may therefore provide fundamental insights into the association between EV function and the detailed disease progression.”

Fig. S21. The proportion and metabolic EV-iNOS level of macrophage-derived EVs (EVs-M, CD4⁺), neutrophil-derived EVs (EVs-N, CD66b⁺) and epithelial cell-derived EVs (EVs-E, EpCAM⁺) among EVs isolated from BALF of a pneumonia patient. EV proportion was identified by EV-based flow cytometry and each proportion was calculated relative to total CD63⁺ EVs. Metabolic EV-iNOS level was measured following an immunobeads-based pull down assay.

Reviewer #4:

The authors have developed a novel methodology for the rapid profiling of EV-iNOS activity, which they have shown to be more sensitive than existing ELISA technologies for measuring EV metabolic activity. Using the developed assay, they quantify iNOS activity in macrophage-derived EVs, which is reflective of the metabolic state of the macrophage and its inflammatory state. They then successfully show differences in macrophage-EV metabolic activity in individuals with pneumonia and healthy counterparts.

Thus, the authors provide promising evidence for the use of EV-iNOS activity as a potential biomarker for lung inflammation, using this novel EV assay. The paper is well-written and concise, but more detail is needed in the EV methodology sections for readers to be able to replicate the results and to meet MISEV standards.

RESPONSE: We greatly appreciate the reviewer's positive recommendation and thoughtful comments. Please find the following specific points.

Q1. Lines 340-346 – It is not clear in the methods whether EV characterisation was performed for BALF-derived EVs. Please indicate which technologies were performed to confirm the presence and characterisation of EVs from the BALF samples.

R1. Thank you very much for this kind suggestion. We have clarified in the Methods section that both TEM, NTA and Western blot analysis were performed for the BALF-derived EVs. Actually, we similarly performed NTA to determine particle size distribution and concentration, TEM to visualize the EV morphology, and Western blotting to detect EV-associated protein markers. The relevant details have been added to the revised Methods section. The revised Main Text is shown as follow:

“For EVs collected from both cell culture supernatants and BALF samples, TEM, NTA, and Western blot analyses were performed to confirm the EV characteristics.”

Q2. General – There is evidence of TEM and NTA profiling of EVs, but according to MISEV 2023 guidelines, EV tetraspanin measurements are recommended when reporting EV data. Unless there is a convincing argument against this, please present data for CD9/CD63/CD81 expression on the EVs of interest.

R2. Thank you very much for pointing this out. We have supplemented Western blot data to demonstrate the expression of the characteristic marker of CD9, CD63, and CD81 in both cell-derived and BALF-derived EVs. The supplemented data have been included in the revised manuscript as follows:

Figure 1. (C) TEM image and Western blot analysis of macrophage-derived EVs. scale bar, 100 nm.

Fig. S16 Western blot analysis of protein markers including CD9, CD63, Tsg101 and iNOS from BALF EVs collected from a patient with pneumonia.

Q3. The term ‘rapid assay’ is used throughout, but it is not clear how long the assay actually takes. Please state in the methods or results the approximate the time for the assay.

R3. We appreciate this constructive comment. We have clarified the assay duration in the Methods section. The complete iNOS activity assay, including EV lysis, enzymatic incubation, and electrochemical readout, requires less than 1 hour, which is substantially faster than conventional ELISA method. Specifically, EV lysis takes ~15 min, the enzymatic reaction requires ~30 min, and the electrochemical signal can be obtained within a few minutes. This detail information has been added to the Main Text is as follows:

“Regarding the detection time, EV lysis requires ~15 min, the enzymatic reaction takes ~30 min, and the electrochemical readout could be obtained within a few minutes.”

Q4. Is it a “non-invasive” method for pneumonia treatment evaluation if it requires BALF collection? This needs re-considering.

R4. Thank you very much for pointing this out. We agree with the reviewer that BALF collection might not be a standard “non-invasive” method, especially for pneumonia treatment evaluation. In the revision, we now refer to it as a radiation-free or minimally invasive approach, which is recognized to provide direct window for rapid lung disease evaluation. This phrase has been replaced throughout the manuscript.

Q5. Minor corrections

- *Line 45 – remove ‘etc’.*
- *Line 308 - ‘in vitro’ should be italicised throughout.*
- *Line 330 - State how many BALF samples were used in the study.*
- *Lines 349-350 - State the manufacturer for the ELISA kit and details for the capture antibody.*

R5. Thank the reviewer for carefully check of our manuscript. All these mistakes have been corrected.

- Line 45 – the word “etc” has been removed for clarity and precision.
- Line 308 - ‘in vitro’ has been italicized
- Line 330 - A total 50 BALF samples were employed in the clinical trial, including 32 pneumonia patients and 18 healthy individuals.
- Lines 349-350 - Mouse and human iNOS ELISA kit were purchased from Beijing 4A Biotech (Beijing, China) and Ruixin Biotech (Quanzhou, China), respectively. The capture antibodies were supplied as part of the commercial ELISA kits and were not individually detailed by the manufacturer.

Reviewer #5:

The manuscript presents an electrochemical assay for measuring the activity of inducible nitric oxide synthase (iNOS) available on extra cellular vesicles by analyzing its enzymatic product, nitric acid. The manuscript is interesting and well written and can be published after addressing the following comments.

RESPONSE: We greatly appreciate the reviewer's positive recommendation and thoughtful comments. Please find the following specific points.

Q1. The clinical sensitivity and specificity of the assay should be determined and quoted against the primary method used for diagnosing pneumonia and not just ELISA. The primary diagnostic method should be discussed.

R1. Thank you very much for this valuable comment. We agree that the clinical sensitivity and specificity of the assay should be evaluated against the primary diagnostic method for pneumonia, rather than solely compared with ELISA. In this study, the classification of pneumonia patients and healthy controls was determined based on an integrated clinical diagnosis, including chest X-ray or CT imaging together with clinical symptoms, in accordance with routine clinical practice. This integrated clinical assessment served as the reference standard for all diagnostic performance analyses. Based on this clinical reference, receiver operating characteristic (ROC) analysis was performed to evaluate sensitivity and specificity. As shown in Figure 4F, the ACTIVITY assay demonstrated the highest diagnostic performance, with an AUC of 0.97, indicating high sensitivity and specificity for distinguishing pneumonia patients from controls. In contrast, EV-based iNOS ELISA showed substantially lower discriminatory ability (AUC = 0.67), underscoring the limitation of protein abundance-based EV measurements. Following the reviewer's suggestion, we also included serum C-reactive protein (CRP) as a representative systemic inflammatory marker for comparison. ROC analysis for CRP (Figure 4F) yielded an AUC of 0.91. While CRP provides useful supportive information, it reflects systemic inflammation and is not specific to lung pathology. In contrast, the ACTIVITY assay measures local inflammatory activity in the lung via BALF-derived EV-associated metabolic function, which likely accounts for its superior diagnostic performance relative to CRP. Moreover, to further strengthen the reliability of the clinical evaluation, we expanded the number of clinical samples from the original 29 to 50. These newly acquired data consistently confirm the capability and robustness of the ACTIVITY method in distinguishing pneumonia patients from healthy individuals. We have incorporated this clarification and the corresponding discussion in the revised manuscript.

The revised Main Text is as follow:

“BALF samples were first centrifuged to remove impurities, and the resulting EVs were subsequently collected, lysed, and subjected to ACTIVITY assaying to determine the metabolic EVs-iNOS levels. We first analyzed EVs from BALF samples collected from 32 pneumonia patients and 18 healthy controls, with patient groups defined according to the integrated clinical diagnosis (see supplementary information for clinical information for these participants).”

Figure 4. Clinical application of the ACTIVITY strategy for pneumonia diagnostics. (D) The ACTIVITY (top) and ELISA (bottom) readouts of clinical samples, including 32 pneumonia patients and 18 healthy individuals. The signal intensities were averaged over triplicate measurements of each sample and normalized by min-max normalization. (E) Statistical analysis of the metabolic EV-iNOS levels in pneumonia patients and healthy controls. (F) ROC curves for the metabolic EV-based ACTIVITY method (red line), the iNOS-based ELISA method (black line) and the serum parameter CRP (gray line). (G) Left: Representative CT images of pneumonia patient before and after clinical treatment. Right: Evaluation of the metabolic EV-iNOS levels in BALF samples from pneumonia patients (n=4) before and after treatment. Both the ACTIVITY and ELISA data were normalized to the EV numbers.

Q2. The advantage of using WS₂ to measure the sensor response over other carbon-based or catalytic metallic electrodes should be demonstrated using experiments.

R2. Thank you very much for this comment. Regarding the comments on the comparison of WS₂ over other carbon-based or catalytic metallic electrodes, we would appreciate having this opportunity to clarify this point. Actually, the main focus of our work is to establish the practical relevance of EV metabolic activity in lung inflammation, which is achieved through the sensitive detection of EV-generated NO. With the integration of biocatalysis and electrocatalysis for dual amplification, the electrochemical readout of NO was achieved by leveraging the catalytic properties of WS₂ QDs. This metabolic EV-iNOS activity-based method shows higher sensitivity than the conventional iNOS protein-based ELISA, facilitating the detection of lung inflammation with BALF samples. As a result, this novel metabolic activity-based EV liquid biopsy method shows superior performance in disease diagnosis, providing a practical alternative to traditional diagnostic techniques. As mentioned above, the comparative advantage of WS₂ over other materials might not be the central theme of our study. Due to challenges in preparation and acquisition of other carbon-based materials, we have referred to relevant literature that demonstrates the catalytic properties of WS₂ QDs in similar applications and found that reduced graphene oxide (rGO) is among the effective carbon-based electrocatalysts for related electrochemical reactions. Therefore, we choose the commercial rGO materials under optimized conditions and performed a direct comparison under same conditions. The results showed that our WS₂-based electrode exhibits better electrocatalytic performance than rGO, further

supporting the suitability of WS₂ as the electrocatalytic component in our platform. The revised Main Text is as follow:

“Also, these modified electrodes exhibited better electrocatalytic performance than the generally used carbon-based electrodes such as reduced graphene oxide-modified electrode (Fig. S5).”

Fig. S5. Cyclic voltammograms of WS₂ QDs (highly defective) and rGO modified electrodes in PBS electrolyte with 0.18 mM NO.

Q3. The normalization used in the different figures is not clear to me, for example, it is written that the CD86/CD206 is used in Figure 3G but it is not clear how other signals have been normalized.

a. What is the justification for using CD86/CD206?

b. How are the electrochemical signals normalized?

c. How are each one of the signals normalized, this should be included in the figure captions and justified in the manuscript

R3. We thank the reviewer for pointing out the need for clarification regarding the normalization procedures in the figures.

a) To confirm macrophage polarization states under different treatment conditions, we performed flow cytometry analysis using the classical surface markers CD86 for M1 phenotype and CD206 for M2 phenotype. The results revealed a clear shift toward higher expression of CD86 and lower CD206 in cell populations in the LPS- and BLZ945-treated groups, consistent with M1-like inflammatory phenotypes. The CD86/CD206 ratio was used to characterize the M1/M2 polarization as a referential indicator of polarization status, supporting the interpretation of EV-associated metabolic activity. The revised Main Text is as follow: “The polarization of macrophages was initially assessed by flow cytometry (Fig. S10). Given that CD86 and CD206 are characteristic markers of M1 and M2 macrophages phenotypes, respectively, the intracellular expression ratio of CD86/CD206 was used as a referential indicator for macrophage polarization.”

b) The electrochemical readouts for clinical samples were generally normalized by min-max normalization after the background subtraction. We have clarified in the Main Text as follow:

“The electrochemical readouts for metabolic EVs-iNOS were obtained after background subtraction, which accounted for the possible interference of lysis buffer and enzyme substrate in PBS. A min-max normalization was subsequently used unless otherwise stated.”

c) We have added detailed descriptions of the normalization procedures to the captions of the relevant figures.

Reviewer #1:

In this revision, the authors have answered some of my comments; however, there are some major issues that need to be addressed.

Q1. In cellular iNOS knockout assay, besides the GFP expression, more direct results (iNOS protein expression) should be provided using western blotting. Also, the effect of cellular iNOS overexpression on EV properties is needed.

R1. Thank you very much for this constructive suggestion. As suggested, we have supplemented the western blot analysis of iNOS protein expression, which has now been included as Figure S15 in the supporting information. The results confirm the absence of iNOS protein expression in these transfected RAW 264.7 cells. Together with the originally provided GFP expression data, these results demonstrate that the iNOS knockout model was successfully established using the proposed CRISPR/Cas9 system. While the effect of cellular iNOS overexpression on EV properties, in fact, is the central point of our manuscript. In the main text, we used LPS stimulation to polarize the macrophage into a pro-inflammatory state, which is known to induce endogenous iNOS upregulation. The corresponding EV properties, specifically the metabolic EVs-iNOS levels, were quantified using our proposed ACTIVITY method, facilitating this metabolic EVs-based profiling of the macrophage polarization.

Fig. S15. Western blot analysis of wild-type RAW264.7 macrophages and those transfected for iNOS knockout.

Q2. The concerns about the induction method of macrophages used in this study still exist. If the cell models cannot correctly represent the changes in diseased state, it may result in misleading or false conclusions.

R2. We appreciate the reviewer's concern regarding the macrophage induction strategy and its physiological relevance. We acknowledge that our original use of the M1/M2 terminology was not sufficiently precise. In this study, LPS was used to modulate macrophages toward a pro-inflammatory activation state under controlled conditions, which has been employed in various studies to investigate disease-associated inflammatory states.^[1-4] Accordingly, to avoid overinterpretation of macrophage polarization states, we have revised the expression throughout the manuscript to replace "M1/M2" with "pro-inflammatory" and "anti-inflammatory". This modification may more accurately reflect the in vitro polarization conditions.

To further confirm the validity of our induction system, we have now supplemented additional Western blot

analysis showing that LPS stimulation markedly upregulates iNOS protein expression (Fig. S10), confirming successful induction of a pro-inflammatory phenotype. Importantly, our conclusions are based on multiple complementary models, including RAW 264.7 macrophages, primary mouse macrophage, as well as primary human macrophage. The consistency of results across these complementary models strengthens the physiological relevance and robustness of our conclusions.

Furthermore,

Fig. S10. Western blot analysis of different polarized RAW264.7 macrophages.

References:

1. Xing, Y., Wang, M., Zhang, F. et al. Lysosomes finely control macrophage inflammatory function via regulating the release of lysosomal Fe²⁺ through TRPML1 channel. *Nat. Commun.* 16, 985 (2025).
2. Casey, A.M., Ryan, D.G., Prag, H.A. et al. Pro-inflammatory macrophages produce mitochondria-derived superoxide by reverse electron transport at complex I that regulates IL-1 β release during NLRP3 inflammasome activation. *Nat. Metab.* 7, 493–507 (2025).
3. Fang, F., Wang, E., Yang, H. et al. Reprogramming mitochondrial metabolism and epigenetics of macrophages via miR-10a liposomes for atherosclerosis therapy. *Nat. Commun.* 16, 9117 (2025).
4. Zhou, Q., Gao, J., Wu, G. et al. Adipose progenitor cell-derived extracellular vesicles suppress macrophage M1 program to alleviate midlife obesity. *Nat. Commun.* 16, 2743 (2025).

Q3. *The authors claimed that “IL-4 pretreatment was applied to establish an initial M2-like anti-inflammatory state, thereby providing a defined baseline from which dynamic phenotypic changes could be examined following inflammatory stimulation”. However, the in vivo IL-4 levels in physiologic state are relatively low, and previous papers have reported that the IL-4-to-LPS induction can induce non-canonical proinflammatory M2^{INF} macrophages (e.g., PMID: 37149865) that display a different phenotype compared to M1.*

R3. Thank you very much for pointing this out. We fully acknowledge that physiological IL-4 levels in vivo are relatively low and that sequential IL-4 and LPS stimulation can generate macrophage phenotypes that do not strictly correspond to the classical M1/M2 paradigm. In our study, IL-4 pretreatment was not intended to precisely replicate physiological cytokine concentrations in vivo, but rather to establish a controlled anti-inflammatory -like baseline state in vitro. This experimental design allowed us to examine dynamic phenotypic transitions upon subsequent inflammatory stimulation under defined and reproducible

conditions. We agree that macrophage polarization represents a spectrum rather than discrete states. Therefore, to avoid oversimplification, we have revised the manuscript to consistently use the term “anti-inflammatory” instead of “M2,” and we have clarified that the IL-4→LPS sequential stimulation model may produce a mixed or transitional phenotype rather than a canonical M1 state. Importantly, our conclusions are based on the relative changes in phenotype-associated markers and functional readouts, rather than on the assumption of strictly binary polarization states.

Q4. The authors also claimed that “BLZ945 was used as a phenotype-modulating perturbation based on prior evidence that inhibition of CSF-1R signaling by BLZ945 suppresses M2 polarization and shifts macrophage phenotypes toward a more pro-inflammatory state in vivo. However, in this study, the in vitro culture medium does not contain its ligand (CSF-1), so what’s the exact role of the CSF-1R inhibitor?”

R4. Thank you very much for pointing this out. In our present study, BLZ945 was employed as a pharmacological tool to modulate macrophage polarization states. Our primary objective was to determine whether altering CSF-1R activity would influence macrophage phenotypic polarization under inflammatory stimulation, rather than to investigate the precise ligand-receptor dynamics in vitro. Although exogenous CSF-1 was not supplemented in the culture medium, macrophages constitutively express CSF-1R, and basal receptor activity may still exist. Therefore, inhibition of CSF-1R signaling can functionally influence macrophage polarization status. Importantly, our data demonstrate that BLZ945 treatment significantly alters polarization-associated markers and functional readouts, supporting its role as a phenotype-modulating perturbation in this experimental design. To avoid overinterpretation, we have revised the manuscript to clarify that BLZ945 was used as a macrophage phenotype-modulating agent rather than as a tool to specifically interrogate ligand-driven CSF-1R signaling mechanisms.

Q5. Because of the biological difference between the cell lines (RAW 264.7 cells) and the primary cells, the key finding of this study (including above) should be validated using primary mouse or human macrophages.

R5. Thank you very much for this constructive suggestion. We fully agree that the validation in primary macrophages is essential due to the biological difference between the cell lines (RAW 264.7 cells) and the primary cells. As suggested, we have now supplemented our study with additional experiments using primary mouse macrophages, which were collected from mouse peritoneal (Figure 3O). The results obtained from mouse peritoneal macrophage are consistent with those observed in RAW 264.7 cells, demonstrating the reliable regulatory trend and functional effects described in the revised manuscript. Moreover, as included in our previous revision, we have provided validation data using primary human monocyte-derived macrophages (Figure 4A in this revised Main Text). Taken together, the newly supplemented data from primary mouse macrophages, along with the previously included results from primary human macrophages, consistently support our key conclusions. These additional experiments substantially strengthen the biological relevance and robustness of our study.

Figure 3. (O) Schematic illustration of the collection and polarization of mouse primary macrophage and corresponding ACTIVITY readout of metabolic EVs-iNOS levels.